# A COGNITIVE-INSPIRED MULTI-MODULE ARCHITECTURE FOR CONTINUAL LEARNING

## ABSTRACT

Artificial neural networks (ANNs) exhibit a narrow scope of expertise on stationary independent data. However, data in the real world is continuous and dynamic, and ANNs must adapt to novel scenarios while also retaining the learned knowledge to become lifelong learners. The ability of humans to excel at these tasks can be attributed to multiple factors ranging from cognitive computational structures, cognitive biases, and the multi-memory systems in the brain. We incorporate key concepts from each of these to design a cognitive-inspired continual learning method. *Cognitive Continual Learner (CCL)* includes multiple modules, implicit and explicit knowledge representation dichotomy, inductive bias, and a multi-memory system. CCL shows improvement across different settings and also shows a reduced task recency bias. To test versatility of continual learning methods on a challenging distribution shift, we introduce a novel domain-incremental dataset *DN4IL*. In addition to improved performance on existing benchmarks, CCL also demonstrates superior performance on this dataset.[1]

## 1 INTRODUCTION

Deep learning has seen rapid progress in recent years, and supervised learning agents have achieved superior performance in perception tasks. However, unlike a supervised setting, where data is static, and independent and identically distributed, real-world data is changing dynamically. Continual learning (CL) aims at learning multiple tasks when data is streamed sequentially (Parisi et al., 2019). This is crucial in real-world deployment settings, as the model needs to adapt quickly to novel data (plasticity), while also retaining previously learned knowledge (stability). Artificial neural networks (ANN), however, are still not effective continual learners as they often fail to generalize to small changes in distribution and also suffer from forgetting old information when presented with new data (catastrophic forgetting)(McCloskey & Cohen, 1989).

Humans, on the other hand, show a better ability to acquire new skills while also retaining previously learned skills to a greater extent. This intelligence can be attributed to different factors in human cognition. Multiple theories have been proposed to formulate an overall cognitive architecture, which is a broad domain-generic cognitive computation model that captures the essential structure and process of the mind. Some of these theories hypothesize that, instead of a single standalone module, multiple modules in the brain share information to excel at a particular task. CLARION (Connectionist learning with rule induction online) (Sun & Franklin, 2007) is one such theory that postulates an integrative cognitive architecture, consisting of a number of distinct subsystems. It predicates a dual representational structure (Chaiken & Trope, 1999), where the top level encodes conscious explicit knowledge, while the other encodes indirect implicit information. The two systems interact, share knowledge, and cooperate in solving tasks. Delving into these underlying architectures and formulating a new design can help in the quest of building intelligent agents.

Multiple modules can be instituted instead of a single feedforward network. An explicit module that learns from the standard visual input and an implicit module that shares indirect contextual knowledge. The implicit module can be further divided into more sub-modules, each providing different information. Inductive biases and semantic memories can act as different kinds of implicit knowledge. Inductive biases are pre-stored templates or knowledge that provide some meaningful disposition toward adapting to the continuously evolving world (Chollet, 2019). Furthermore,

---

[1]Code and the *DN4IL* dataset will be made accessible upon acceptance.

theories (Kumaran et al., 2016) postulate that after rapidly learning information, a gradual consolidation of knowledge transpires in the brain for slow learning of structured information. Thus, the new design incorporates multiple concepts of cognition architectures, the dichotomy of implicit and explicit representations, inductive biases, and multi-memory systems theory.

To this end, we propose *Cognitive Continual Learner* (CCL), a multi-module architecture for CL. The explicit working module processes the standard input data. Two different sub-modules are introduced for the implicit module. The inductive bias learner embeds relevant prior information, and as networks are shown to be biased toward textural information (unlike humans that are more biased toward global semantics)(Geirhos et al., 2018), we propose to utilize the global shape information as the prior. Shape is already present in the visual data but in an indirect way, and extracting this implicit information and sharing with the explicit module will help to learn more generic and high-level representations. Further, to emulate the consolidation of information in the slow-fast multi-memory system, a gradual accumulation of knowledge from the explicit working module is embedded in the second semantic memory sub-module. We show that interacting and leveraging information between these modules can help alleviate catastrophic forgetting while also increasing the robustness to distribution shift.

CCL achieves superior performance across all CL settings on various datasets. CCL outperforms the SOTA CL methods on Seq-CIFAR10, Seq-CIFAR100 in the class incremental settings. Furthermore, in more realistic general class incremental settings where the task boundary is blurry and classes are not disjoint, CCL shows significant gains. The addition of inductive bias and semantic memory helps to achieve a better balance between the plasticity-stability trade-off. The prior in the form of shape helps produce generic representations, and this results in CCL exhibiting a reduced task-recency bias. Furthermore, CCL also shows higher robustness against natural corruptions. Finally, to test the capability of the CL methods against distribution shift, we introduce a domain incremental learning dataset, *DN4IL*, which is a carefully designed subset of the DomainNet dataset (Peng et al., 2019). CCL shows considerable robustness across all domains on these challenging data, thus establishing the efficacy of our cognitive-inspired CL architecture. Our contributions are as follows:

- *Cognitive Continual Learner (CCL)*, a novel method that incorporates aspects of cognitive architectures, multi-memory systems, and inductive bias into the CL framework.
- Introducing *DN4IL*, a challenging domain incremental learning dataset for CL.
- Benchmarks across different CL settings: class incremental, task incremental, generalized class incremental, and domain incremental learning.
- Analyses on the plasticity-stability trade-off, task recency bias, and robustness to natural corruptions.

## 2 METHODOLOGY

### 2.1 COGNITIVE ARCHITECTURES

Cognitive architectures refer to computational models that encapsulate the overall structure of the cognitive process in the brain. The underlying infrastructure of such a model can be leveraged to develop better intelligent systems. Global workspace theory (GWT) (Juliani et al., 2022) postulates that human cognition is composed of a multitude of special-purpose processors and is not a single standalone module. Different sub-modules might encode different contextual information which, when activated, can transfer knowledge to the conscious central workspace to influence and help make better decisions. Furthermore, CLARION (Sun & Franklin, 2007) posits a dual-system cognitive architecture with two levels of knowledge representation. The explicit module encodes direct knowledge that is externally accessible. The implicit module encodes indirect knowledge that is not directly accessible, but can be obtained through some intermediate interpretive or transformational steps. These two modules interact with each other by transferring knowledge between each other.

Inspired by these theories, we formulate a method that incorporates some of the key aspects of cognitive architecture into the CL method. A working module, which encodes the direct sensory data, forms the explicit module. A second module that encodes indirect and interpretive information forms the implicit module. The implicit module further includes multiple sub-modules to encode different types of knowledge.

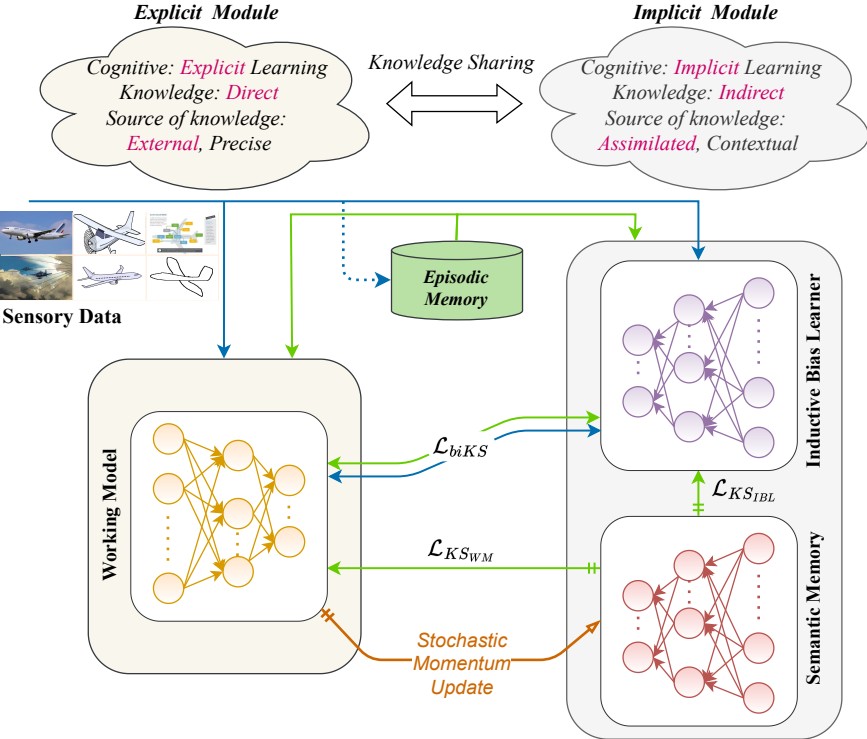

Figure 1: Schematic of *Cognitive Continual Learner (CCL)*. The working model in explicit module learns direct sensory data. Within the implicit module, the inductive bias learner encodes the prior shape knowledge and the semantic memory consolidates information from the explicit module.

## 2.2 INDUCTIVE BIAS

The sub-modules in the implicit module need to encapsulate implicit information that can provide more contextual and high-level supervision. One of such knowledge can be prior knowledge or inductive bias. Inductive biases are prestored templates that exist implicitly even in earlier stages of the human brain (Pearl & Mackenzie, 2018). For instance, cognitive inductive bias may be one of the reasons why humans can focus on the global semantics of objects to make predictions. ANNs, on the other hand, are more prone to rely on local cues and textures (Geirhos et al., 2018). Global semantics or shape information already exists in the visual data, but in an indirect way. Hence, we utilize shape as indirect information in the implicit module. The sub-module uses a transformation step to extract the shape and share this inductive bias with the working module. As the standard (RGB) image and its shape counterpart can be viewed as different perspectives/modalities of the same data, ensuring that the representation of one modality is consistent with the other increases robustness to spurious correlations that might exist in only one of them.

## 2.3 MULTI MEMORY SYSTEM

Moreover, many theories have postulated that an intelligent agent must possess deferentially specialized learning memory systems (Kumaran et al., 2016). While one system rapidly learns the individual experience, the other gradually assimilates the knowledge. To emulate this behavior, we establish a second sub-module that slowly consolidates the knowledge from the working module.

## 2.4 FORMULATION

To this end, we propose a novel method *Cognitive Continual Learner (CCL)*, which incorporates all these concepts into the CL paradigm. CCL consists of two modules, the explicit module and the implicit module. The explicit module has a single working model and processes the incoming

direct visual data. The implicit module further consists of two sub-modules, namely the inductive bias learner and the semantic memory. They share relevant contextual information and assimilated knowledge with the explicit module, respectively. Figure 1 shows the overall architecture.

In the implicit module, semantic memory $N_{SM}$, consolidates knowledge at stochastic intervals from the working model $N_{WM}$, in the explicit module. The other sub-module, the inductive bias learner $N_{IBL}$, processes the data and extracts the shape information (Section B). $N_{WM}$ processes the RGB data, $N_{SM}$ consolidates the information from the working module at an update frequency in a stochastic manner, and $N_{IBL}$ learns from the shape data. $f$ represents the combination of the encoder and the classifier, and $\theta_{WM}$, $\theta_{SM}$, and $\theta_{IBL}$ are the parameters of the three networks.

A CL classification consists of a sequence of $T$ tasks and, during each task $t \in 1, 2...T$, samples $x_c$ and their corresponding labels $y_c$ are drawn from the current task data $D_t$. Furthermore, for each subsequent task, a random batch of exemplars is sampled from episodic memory $B$ as $x_b$. An inductive bias (shape) filter is applied to generate shape samples, $x_{c_s} = \mathbb{IB}(x_c)$ and $x_{b_s} = \mathbb{IB}(x_b)$. Reservoir sampling (Vitter, 1985) is incorporated to replay previous samples. Each of the networks $N_{WM}$ and $N_{IBL}$ learns in its own modality with supervised cross-entropy loss on both the current samples and the buffer samples:

$$\mathcal{L}_{Sup_{WM}} = \mathcal{L}_{CE}(f(x_c; \theta_{WM}), y_c) + \mathcal{L}_{CE}(f(x_b; \theta_{WM}), y_b) \tag{1}$$

$$\mathcal{L}_{Sup_{IBL}} = \mathcal{L}_{CE}(f(x_{c_s}; \theta_{IBL}), y_c) + \mathcal{L}_{CE}(f(x_{b_s}; \theta_{IBL}), y_b) \tag{2}$$

The Knowledge Sharing (KS) objectives are designed to transfer and share information between all modules. KS occurs for current samples and buffered samples. We employ the mean squared error as the objective function for all KS losses. To provide shape supervision to the working model and vice versa, a bidirectional decision space similarity constraint ($\mathcal{L}_{biKS}$) is enforced to align the features of the two modules.

$$\mathcal{L}_{biKS} = \mathop{\mathbb{E}}_{x \sim D_t \cup B} \|f(x_s; \theta_{IBL}) - f(x; \theta_{WM})\|_2^2 \tag{3}$$

The consolidated structural information in semantic memory is transferred to both the working model and the inductive bias learner by aligning the output space on the buffer samples, which further helps in information retention. The loss functions $\mathcal{L}_{KS_{WM}}$ and $\mathcal{L}_{KS_{IBL}}$ are as follows;

$$\mathcal{L}_{KS_{WM}} = \mathop{\mathbb{E}}_{x_b \sim B} \|f(x_b; \theta_{SM}) - f(x_b; \theta_{WM})\|_2^2 \tag{4}$$

$$\mathcal{L}_{KS_{IBL}} = \mathop{\mathbb{E}}_{x_b \sim B} \|f(x_b; \theta_{SM}) - f(x_{b_s}; \theta_{IBL})\|_2^2 \tag{5}$$

Thus, the overall loss functions for the working model and the inductive bias learner are as follows;

$$\mathcal{L}_{WM} = \mathcal{L}_{Sup_{WM}} + \lambda \mathcal{L}_{biKS} + \lambda' \mathcal{L}_{KS_{WM}} \tag{6}$$

$$\mathcal{L}_{IBL} = \mathcal{L}_{Sup_{IBL}} + \gamma \mathcal{L}_{biKS} + \gamma' \mathcal{L}_{KS_{IBL}} \tag{7}$$

The semantic memory of the implicit module is updated with a stochastic momentum update (SMU) of the weights of the working model at rate $r$ with a decay factor of $d$,

$$\theta_{SM} = d \cdot \theta_{SM} + (1 - d) \cdot \theta_{WM} \text{ if } s \sim U(0, 1) < r \tag{8}$$

More details are provided in Algorithm 2. Note that we use semantic memory ($\theta_{SM}$) for inference, as it contains consolidated knowledge across all tasks.

## 3 EXPERIMENTAL SETTINGS

ResNet-18 (He et al., 2016) architecture is used for all experiments. All networks are trained using the SGD optimizer with standard augmentations of random crop and random horizontal flip. The different hyperparameters, tuned per dataset, are provided in E. The different CL settings are explained in detail in Section D. We consider CLass-IL, Domain-IL and also report the Task-IL settings. Seq-CIFAR10 and Seq-CIFAR100 (Krizhevsky et al., 2009) for the class incremental learning (Class-IL) settings, which are divided into 5 tasks each. As an addition to Class-IL, we also consider and evaluate General Class-IL (GCIL) (Mi et al., 2020) on CIFAR100 dataset. For the domain incremental learning (Domain-IL), we propose a novel dataset, *DN4IL*.

Table 1: Comparison of different methods on standard CL benchmarks (Class-IL, Task-IL and GCIL settings). CCL shows a consistent improvement over all methods for both buffer sizes.

| $|\mathcal{B}|$ | Method | Seq-CIFAR10 | | Seq-CIFAR100 | | GCIL-CIFAR100 | |
|---|---|---|---|---|---|---|---|
| | | Class-IL | Task-IL | Class-IL | Task-IL | Uniform | Longtail |
| - | JOINT | $92.20_{\pm 0.15}$ | $98.31_{\pm 0.12}$ | $70.62_{\pm 0.64}$ | $86.19_{\pm 0.43}$ | $60.45_{\pm 1.65}$ | $60.10_{\pm 0.42}$ |
| | SGD | $19.62_{\pm 0.05}$ | $61.02_{\pm 3.33}$ | $17.58_{\pm 0.04}$ | $40.46_{\pm 0.99}$ | $10.36_{\pm 0.13}$ | $9.62_{\pm 0.21}$ |
| 200 | ER | $44.79_{\pm 1.86}$ | $91.19_{\pm 0.94}$ | $21.40_{\pm 0.22}$ | $61.36_{\pm 0.39}$ | $16.52_{\pm 0.10}$ | $16.20_{\pm 0.30}$ |
| | DER++ | $64.88_{\pm 1.17}$ | $91.92_{\pm 0.60}$ | $29.60_{\pm 1.14}$ | $62.49_{\pm 0.78}$ | $27.73_{\pm 0.93}$ | $26.48_{\pm 2.04}$ |
| | $Co^2L$ | $65.57_{\pm 1.37}$ | $93.43_{\pm 0.78}$ | $31.90_{\pm 0.38}$ | $55.02_{\pm 0.36}$ | - | - |
| | ER-ACE | $62.08_{\pm 1.44}$ | $92.20_{\pm 0.57}$ | $32.49_{\pm 0.95}$ | $59.77_{\pm 0.31}$ | $27.64_{\pm 0.76}$ | $25.10_{\pm 2.64}$ |
| | CLS-ER[†] | $66.19_{\pm 0.75}$ | $93.90_{\pm 0.60}$ | $43.80_{\pm 1.89}$ | $73.49_{\pm 1.04}$ | $35.88_{\pm 0.41}$ | $35.67_{\pm 0.72}$ |
| | CCL | $\mathbf{70.04_{\pm 1.07}}$ | $\mathbf{94.49_{\pm 0.38}}$ | $\mathbf{46.55_{\pm 1.51}}$ | $\mathbf{76.66_{\pm 0.41}}$ | $\mathbf{39.61_{\pm 0.70}}$ | $\mathbf{38.94_{\pm 0.16}}$ |
| 500 | ER | $57.74_{\pm 0.27}$ | $93.61_{\pm 0.27}$ | $28.02_{\pm 0.31}$ | $68.23_{\pm 0.16}$ | $23.62_{\pm 0.66}$ | $22.36_{\pm 1.27}$ |
| | DER++ | $72.70_{\pm 1.36}$ | $93.88_{\pm 0.50}$ | $41.40_{\pm 0.96}$ | $70.61_{\pm 0.11}$ | $35.80_{\pm 0.62}$ | $34.23_{\pm 1.19}$ |
| | $Co^2L$ | $74.26_{\pm 0.77}$ | $95.90_{\pm 0.26}$ | $39.21_{\pm 0.39}$ | $62.98_{\pm 0.58}$ | - | - |
| | ER-ACE | $68.45_{\pm 1.78}$ | $93.47_{\pm 1.00}$ | $40.67_{\pm 0.06}$ | $66.45_{\pm 0.71}$ | $30.14_{\pm 1.11}$ | $31.88_{\pm 0.73}$ |
| | CLS-ER | $75.22_{\pm 0.71}$ | $94.94_{\pm 0.53}$ | $51.40_{\pm 1.00}$ | $78.12_{\pm 0.24}$ | $38.94_{\pm 0.38}$ | $38.79_{\pm 0.67}$ |
| | CCL | $\mathbf{76.20_{\pm 0.70}}$ | $\mathbf{95.95_{\pm 0.14}}$ | $\mathbf{53.23_{\pm 1.62}}$ | $\mathbf{80.12_{\pm 0.18}}$ | $\mathbf{44.25_{\pm 0.21}}$ | $\mathbf{42.75_{\pm 0.18}}$ |

## 4 RESULTS

We provide a comparison of our method with standard baselines and multiple other SOTA CL methods. The lower and upper bounds are reported as SGD (standard training) and JOINT (training all tasks together), respectively. We compare with other rehearsal-based methods in the literature, namely ER, DER (Buzzega et al., 2020), $Co^2L$ (Cha et al., 2021), ER-ACE (Caccia et al., 2021) and CLS-ER (Arani et al., 2021). Table S2 shows the average performance in different settings over three seeds. $Co^2L$ utilizes task boundary information, and therefore the GCIL setting is not applicable. The results are taken from the original papers and, if not available, using the original codes, we conducted a hyperparameter search for the new settings.

CCL achieves the best performance across all datasets in all settings. In the challenging Class-IL setting, we observe a gain of ~50% over DER++, thus showing the efficacy of adding multiple modules to CL. Furthermore, we report improvements of ~6% on both the Seq-CIFAR10 and Seq-CIFAR100 datasets, over CLS-ER, which utilizes two memories in its design. CCL has a single semantic memory, and the additional boost is procured by prior knowledge from the inductive bias learner. Improvement is prominent even when the memory budget is low (200 buffer size). GCIL represents a more realistic setting, as the task boundaries are blurry and classes can reappear and overlap in any task. GCIL-Longtail version also introduces an imbalance in the sample distribution. CCL shows a significant improvement on both versions of GCIL-CIFAR100. Shape information from the inductive bias learner offers the global high-level context, which helps in producing generic representations that are not biased towards learning only the current task at hand. Furthermore, sharing of the knowledge that has been assimilated through the appearance of overlapping classes through the training scheme, further facilities learning in this general setting. The overall results indicate that the dual knowledge sharing between the explicit working module and the implicit inductive bias and semantic memory modules enables both better adaptation to new tasks and information retention.

## 5 DOMAIN INCREMENTAL LEARNING

Intelligent agents deployed in real-world applications need to maintain consistent performance through changes in the data and environment. Domain-IL aims to assess the robustness of the CL methods to the distribution shift. In Domain-IL, the classes in each task remain the same, but the input distribution changes, and this makes for a more plausible use case for evaluation. However, the datasets used in the literature do not fully reflect this setting. For instance, the most common

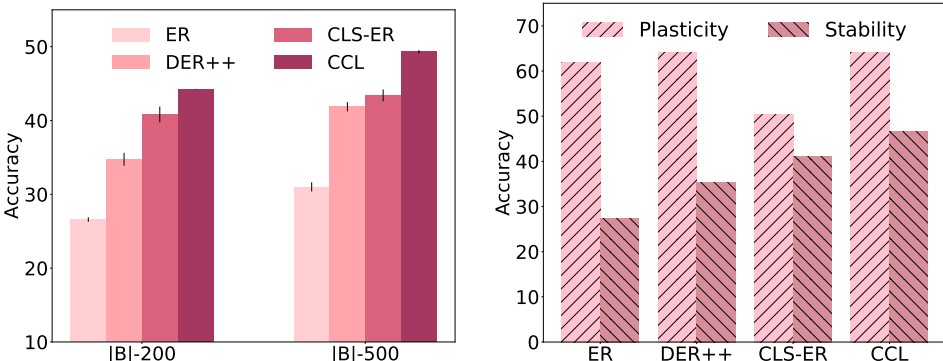

Figure 2: Accuracy (left) and plasticity-stability analysis (right) on *DN4IL* dataset. CCL substantially outperforms other methods and demonstrates a better plasticity-stability trade-off.

datasets used in the literature are different variations (Rotated and Permuted) of the MNIST dataset (LeCun et al., 1998). MNIST is a simple dataset, usually evaluated on MLP networks, and its variations do not reflect the real-world distribution shift challenges that a CL method faces. As is evident from the different CL methods in the literature, the improvement in performance has been saturated on all variants of MNIST. Farquhar & Gal (2018) propose fundamental desiderata for CL evaluations and datasets based on real-world use cases. One of the criteria is to possess cross-task resemblances, which Permuted-MNIST clearly violates. Thus, a different dataset is needed to test the overall capability of a CL method to handle the distributional shift.

### 5.1 DN4IL DATASET

To this end, we propose *DN4IL* (DomainNet for Domain-IL), which is a well-crafted subset of the standard DomainNet dataset (Peng et al., 2019), used in domain adaptation. DomainNet consists of common objects in six different domains - real, clipart, infograph, painting, quickdraw, and sketch. The original DomainNet consists of 59k samples with 345 classes in each domain. The classes have redundancy, and moreover, evaluating the whole dataset can be computationally expensive in a CL setting. Considering different criteria such as the relevance of classes, uniform sample distribution, computational complexity, and ease of benchmarking for CL, we create the version *DN4IL*, which is tailor-made for continual learning.

All classes were grouped into semantically similar supercategories. Out of these, a subset of classes was selected that had relevance to domain shift while also having maximum overlap with other standard datasets such as CIFAR-10 and CIFAR-100, as this can facilitate in performing out-of-distribution analyses. 20 supercategories were chosen with 5 classes each (resulting in a total of 100 classes). In addition, to provide a balanced dataset, we performed a class-wise sampling. First, we sample images per class in each supercategory and maintain class balance. Second, we choose samples per domain, so that it results in a dataset that has a near-uniform distribution across all classes and domains. The final dataset *DN4IL* is succinct, more balanced, and more computationally efficient for benchmarking, thus facilitating research in CL. Additionally, the new dataset deems more plausible for real-world settings and also adheres to all the evaluation desiderata by (Farquhar & Gal, 2018). The challenging distribution shift between domains provides an apt dataset to test the capability of CL methods in the Domain-IL setting. More details, statistics, and visual examples of this crafted dataset are provided in Section I.

### 5.2 DN4IL PERFORMANCE

Figure 2 (left) reports the results on *DN4IL* for two different buffer sizes (Values are provided in Table S8). CCL shows a considerable performance gain in the average accuracy across all domains and can be primarily attributed to the supervision from the shape data. Standard networks tend to exhibit texture bias and learn background or spurious cues (Geirhos et al., 2018) that result in performance degradation when the distribution changes. Learning global shape information of objects,

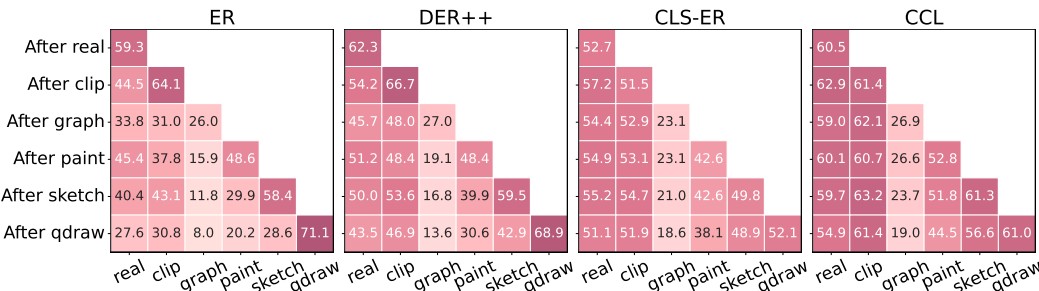

Figure 3: Task-wise performance on *DN4IL* ($|\mathcal{B}|$=500), where each task represents a domain. CCL shows more retention of old information without compromising much on current accuracy.

on the other hand, helps in learning generic features that can translate well to other distributions. Semantic memory further helps to consolidate information across domains. Maintaining consistent performance to such difficult distribution shift proves beneficial in real-world applications, and the proficiency of CCL in this setting can thus open up new avenues for research in cognition-inspired multi-module architectures.

## 6 ANALYSIS

### 6.1 PLASTICITY-STABILITY TRADE-OFF

Plasticity refers to the capability of a model to learn new tasks, while stability shows how well it can retain old information. The plasticity-stability dilemma is a long-standing problem in CL, which requires an optimal balance between the two. We measure each of these to assess the competence of the CL methods. Plasticity is computed as the average performance of each task when it is first learned (e.g., the accuracy of the network trained on task $T2$, evaluated on the test set of $T2$). Stability is computed as the average performance of all tasks 1:$T$-1, after learning the final task $T$. Figure 2 (right) reports these numbers for the *DN4IL* dataset. As seen, the ER and DER methods exhibit forgetting and show low stability and concentrate only on the newer tasks. CLS-ER shows greater stability, but at the cost of reduced plasticity. However, CCL shows the highest stability while maintaining comparable plasticity. The shape knowledge in CCL helps in learning generic solutions that can translate to new tasks, while the semantic consolidation update at stochastic rates acts as a regularization to maintain stable parameter updates. Thus, CCL strikes a better balance between plasticity and stability.

### 6.2 TASK-WISE PERFORMANCE

The average accuracy across all tasks does not provide a complete measure of the ability of a network to retain old information while learning new tasks. To better represent the plasticity-stability measure, we report the task-wise performance at the end of each task. After training each task, we measure the accuracy on the test set of each of the previous tasks. Figure 3 reports this for all tasks of *DN4IL*. The last row represents the performance of each task after the training is complete. ER and DER++ show performance degradation on earlier tasks, as the model continues training on newer tasks. Both perform well on the last task and display the lowest stability. CCL reports the highest information retention on older tasks, while also maintaining plasticity. For example, the accuracy on the first task (real) reduces to 27.6 on ER after training the 6 tasks (domains), while the CCL maintains the accuracy of 54.9. CLS-ER shows better retention of old information but at the cost of plasticity. The last task on CLS-ER shows lower performance compared to CCL (52.1 vs. 61.0). Similar trend (with more gains) is seen on Seq-CIFAR10 dataset in Appendix Figure S2. The performance of the current task in CCL is relatively lesser and can be attributed to the stochastic update rate of this model.

To shed more light on the performance of each of the modules in CCL, we also provide the performance of the working model and the inductive bias learner, in Appendix Figure S1. The working model shows better plasticity, while CCL (semantic memory) displays better stability. Overall, all

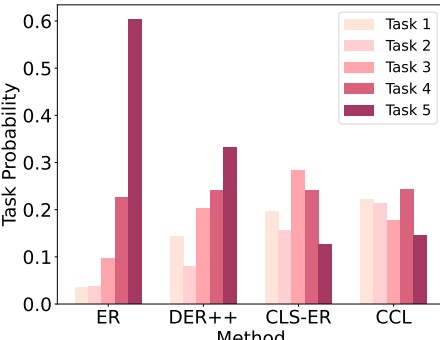 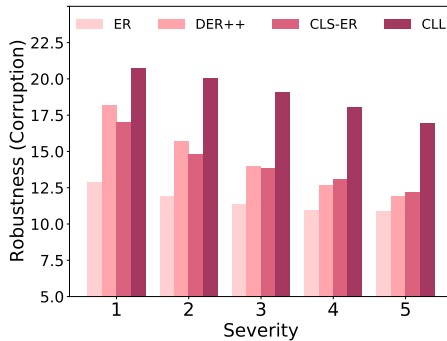

Figure 4: CCL shows reduced task recency bias (left), as well as higher robustness against natural corruption (right) on Seq-CIFAR10 ($|\mathcal{B}|$=200) dataset.

the modules in the proposed approach present unique attributes that improve the learning process and improve performance and reduce catastrophic forgetting.

### 6.3 RECENCY-BIAS ANALYSIS

Recency bias is a behavior in which the model predictions tend to be biased toward the current or the most recent task (Wu et al., 2019). This is undesirable in a CL model, as it results in a biased solution that forgets the old tasks. To this end, after the end of the training, we evaluate the models on the test set (of all tasks) and calculate the probability of predicting each task. The output distribution for each test sample is computed for all classes, and the probabilities are averaged per task.

Figure 4 (left) shows the probabilities for each task on Seq-CIFAR10 dataset. As shown, the ER and DER++ methods tend to incline most of their predictions towards the classes seen in the last task, thus creating a misguided bias. CCL shows a lesser bias compared to both of these baselines. CLS-ER exhibits reduced bias due to the presence of multiple memories, but the distribution is still relatively skewed (w.r.t. probability of 0.2). CCL shows more of a uniform distribution across all tasks. The dual information from the shape data and the consolidated knowledge across tasks helps in breaking away from the Occam's razor pattern of neural networks to default to the easiest solution.

### 6.4 ROBUSTNESS

Lifelong agents, when deployed in real-world settings, must be resistant to various factors, such as lighting conditions, changes in weather, and other effects of digital imaging. Inconsistency in predictions under different conditions might result in undesirable outcomes, especially in safety-critical applications such as autonomous driving. To measure the robustness of the CL method against such natural corruptions, we created a dataset by applying fifteen different corruptions (Table S6), at varying levels of severity (1- least severe to 5- most severe corruption).

The performances on the fifteen corruptions are averaged at each severity level and are shown in Figure 4 (right). CCL outperforms all other techniques at all severity levels. ER, DER++, and CLS-ER show a fast decline in accuracy as severity increases, while CCL maintains stable performance throughout. Implicit shape information provides a different perspective of the same data to the model, which helps to generate high-level, robust representations. CCL, along with improved continual learning performance, also exhibits improved robustness to corruptions, thus proving to be a better candidate for deployment in real-world applications.

### 6.5 ABLATION STUDY

CCL architecture comprises multiple components, each contributing to the efficacy of the method. The explicit module has the working model, and the implicit module has semantic memory (SM) and inductive bias learner (IBL). Disentangling different components in the CCL, can provide more insight into the contribution of each of them to the overall performance.

Table 2: Ablation to analyse the effect of each component of CCL on Seq-CIFAR10 and *DN4IL*.

| SM | IBL | KS (WM↔IBL) | Seq-CIFAR10 | DN4IL |
|----|-----|-------------|-------------|-------|
| ✓ | ✓ | ✓ | $70.04_{\pm1.07}$ | $44.23_{\pm0.05}$ |
| ✓ | ✓ | ✗ | $69.28_{\pm1.34}$ | $40.35_{\pm0.34}$ |
| ✓ | ✗ | - | $69.21_{\pm1.46}$ | $39.76_{\pm0.56}$ |
| ✗ | ✓ | ✓ | $64.61_{\pm1.22}$ | $37.33_{\pm0.01}$ |
| ✗ | ✗ | ✗ | $44.79_{\pm1.86}$ | $26.59_{\pm0.31}$ |

Table 2 reports the ablation study w.r.t to each of these components on both Seq-CIFAR10 and *DN4IL* datasets. Considering the more complex *DN4IL* dataset, the ER accuracy without any of our components is 26.59. Adding cognitive bias (IBL) improves performance by 40%. Shape information plays a prominent role, as the networks need to learn the global semantics of the objects, rather than background or spurious textural information to translate performance across domains. Adding the dual-memory component (SM) shows an increase of approximately 49% over the vanilla baseline. Furthermore, KS between explicit and implicit modules on current experiences also plays a key role in performance gain. Combining both of these cognitive components and, in general, following the multi-module design shows a gain of 66%. A similar trend is seen on Seq-CIFAR10.

## 7 RELATED WORKS

Rehearsal-based approaches, which revisit examples from the past to alleviate catastrophic forgetting, have been effective in challenging CL scenarios (Farquhar & Gal, 2018). Experience Replay (ER) (Riemer et al., 2018) methods use episodic memory to retain previously seen samples for replay purposes. DER++ (Buzzega et al., 2020) adds a consistency loss on logits, in addition to the ER strategy. $CO^2L$ (Cha et al., 2021) uses contrastive learning from the self-supervised learning domain to generate transferable representations. ER-ACE (Caccia et al., 2021) targets the representation drift problem in online CL and develops a technique to use separate losses for current and buffer samples. All of these methods limit the architecture to a single stand-alone network, contrary to the biological workings of the brain. CLS-ER (Arani et al., 2021) proposed a multi-network approach that emulates fast and slow learning systems by using two semantic memories, each aggregating weights at different times. Though CLS-ER utilizes the multi-memory design, sharing of different kinds of knowledge is not leveraged, and hence presents a method with limited scope. CCL digresses from the standard architectures and proposed a multi-module design that is inspired by the cognitive computational architectures. It incorporates multiple sub-modules, each sharing different knowledge to develop an effective continual learner that has better generalization and robustness.

## 8 CONCLUSION

We introduced a novel framework for continual learning which incorporates concepts inspired by cognitive architectures, high-level cognitive biases, and the multi-memory system. Our method, *Cognitive Continual Learner (CCL)*, includes multiple subsystems with dual knowledge representation. CCL designed a dichotomy of explicit and implicit modules in which information is selected, maintained, and shared with each other, to enable better generalization and robustness. CCL outperformed on Seq-CIFAR10 and Seq-CIFAR100 on the Class-IL setting. In addition, it also showed significant gain in the more realistic and challenging GCIL setting. Through different analyses, we showed the better plasticity-stability balance achieved by CCL. Furthermore, shape prior and knowledge consolidation helps to learn more generic solutions, indicated by the reduced task recency bias problem and higher robustness against natural corruptions. Furthermore, we introduced a challenging domain-IL dataset, *DN4IL*, with six disparate domains. The significant improvement of CCL on this complex distribution shift demonstrates the benefits of shape context, which helps the network to converge on a generic solution, rather than a simple texture-biased one. In general, incorporating a design inspired by the cognitive model and sharing information between explicit and implicit inductive bias and implicit semantic memory modules, instead of a standalone network, helps enhance lifelong learning, while also improving generalization and robustness.

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

# A APPENDIX

## A.1 CCL

Figure S1 presents the task-wise performance of all the three networks in the CCL architecture, on *DN4IL* dataset. Semantic memory helps in information retention by maintaining high accuracy on older tasks and is more stable. The performance of the current task is relatively lower than that of the working model and could be due to the stochastic update rate of this model. The working model has better performance on new tasks and is more plastic. Inductive bias leaner is evaluated on the transformed data (shape) and also achieves a balance between plasticity and stability. In general, all modules in our proposed method present unique attributes that improve the learning process by improving performance and reducing catastrophic forgetting.

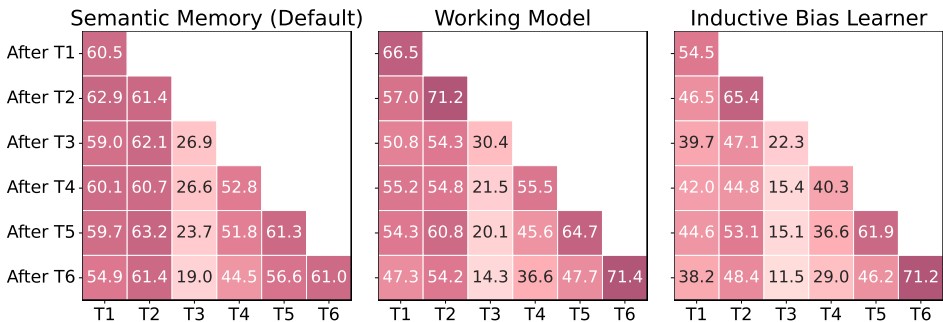

Figure S1: Task probability analysis of all CCL components on *DN4IL* dataset with 500 buffer size. Semantic memory displays better stability while working model displays better plasticity.

Figure S2 presents a similar analysis on the Seq-CIFAR10 dataset. The trend is similar, but the performance gain is much higher on this dataset.

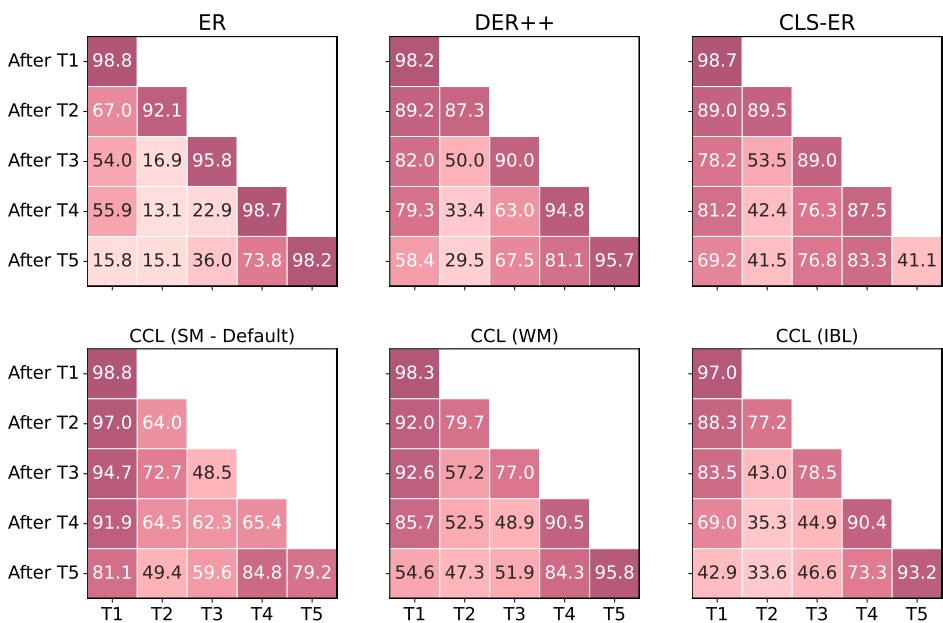

Figure S2: Task probability analysis on Seq-CIFAR10 dataset with 200 buffer size. Semantic memory shows better stability while working model shows better plasticity.

Table S1: Comparison of different methods on standard CL benchmarks (Class-IL, Task-IL settings), including non-ER based methods on Seq-CIFAR10 dataset

| $|\mathcal{B}|$ | Method | Class-IL | Task-IL |
|---|---|---|---|
| - | JOINT | $92.20_{\pm0.15}$ | $98.31_{\pm0.12}$ |
| | SGD | $19.62_{\pm0.05}$ | $61.02_{\pm3.33}$ |
| - | oEWC | $19.49_{\pm0.12}$ | $68.29_{\pm3.92}$ |
| | SI | $19.48_{\pm0.17}$ | $68.05_{\pm5.91}$ |
| | LwF | $19.61_{\pm0.05}$ | $63.29_{\pm2.35}$ |
| | PNN | - | $95.13_{\pm0.72}$ |
| 200 | ER | $44.79_{\pm1.86}$ | $91.19_{\pm0.94}$ |
| | DER++ | $64.88_{\pm1.17}$ | $91.92_{\pm0.60}$ |
| | $Co^2L$ | $65.57_{\pm1.37}$ | $93.43_{\pm0.78}$ |
| | ER-ACE | $62.08_{\pm1.44}$ | $92.20_{\pm0.57}$ |
| | CLS-ER$^\dagger$ | $66.19_{\pm0.75}$ | $93.90_{\pm0.60}$ |
| | CCL | $\mathbf{70.04_{\pm1.07}}$ | $\mathbf{94.49_{\pm0.38}}$ |
| 500 | ER | $57.74_{\pm0.27}$ | $93.61_{\pm0.27}$ |
| | DER++ | $72.70_{\pm1.36}$ | $93.88_{\pm0.50}$ |
| | $Co^2L$ | $74.26_{\pm0.77}$ | $95.90_{\pm0.26}$ |
| | ER-ACE | $68.45_{\pm1.78}$ | $93.47_{\pm1.00}$ |
| | CLS-ER | $75.22_{\pm0.71}$ | $94.94_{\pm0.53}$ |
| | CCL | $\mathbf{76.20_{\pm0.70}}$ | $\mathbf{95.95_{\pm0.14}}$ |

Table S2: Comparison on Seq-CIFAR100 dataset for different tasks on 500 buffer size

| $|\mathcal{B}|$ | Method | 5-Tasks | 10-Tasks | 20-Tasks |
|---|---|---|---|---|
| 500 | ER | $28.02_{\pm0.31}$ | $21.49_{\pm0.47}$ | $16.52_{\pm0.86}$ |
| | DER++ | $41.40_{\pm0.96}$ | $36.20_{\pm0.52}$ | $22.25_{\pm5.87}$ |
| | CCL | $\mathbf{53.23_{\pm1.62}}$ | $\mathbf{41.09_{\pm0.72}}$ | $\mathbf{33.60_{\pm0.25}}$ |

# B    INDUCTIVE BIAS

The shape extraction is performed by applying a filter on the input image. Multiple filters were considered (such as Canny (Ding & Goshtasby, 2001), Prewitt), but the Sobel filter (Sobel & Feldman, 1968) was chosen because it produces a more realistic output by being precise and also smoothing the edges. The overall algorithm is explained in the following.

---
**Algorithm 1** Sobel Algorithm - Shape Extraction

---
**Input:** Input data $x_{rgb}$
1: Up-sample the images to twice the original size: $x_{rgb} = \text{us}(x_{rgb})$
2: Apply Gaussian smoothing to reduce noisy edges: $x_g = \text{Gaussian\_Blur}(x_{rgb}, kernel\_size = 3)$
3: Get Sobel kernels: $S_x = \begin{bmatrix} -1 & 0 & +1 \\ -2 & 0 & +2 \\ -1 & 0 & +1 \end{bmatrix}$ and $S_y = \begin{bmatrix} -1 & -2 & -1 \\ 0 & 0 & 0 \\ +1 & +2 & +1 \end{bmatrix}$
4: Apply Sobel kernels: $x_{dx} = x_g * S_x$ and $x_{dy} = x_g * S_y$
    $*$ : the 2-dimensional convolution operation
5: The edge magnitude: $x_{shape} = \sqrt{x_{dx}^2 + x_{dy}^2}$
6: Down-sample to original image size: $x_{shape} = \text{ds}(x_{shape})$

---

Figure S3 displays few examples of applying the Sobel operator on the original RGB images. The Sobel output is fed to the IBL model.

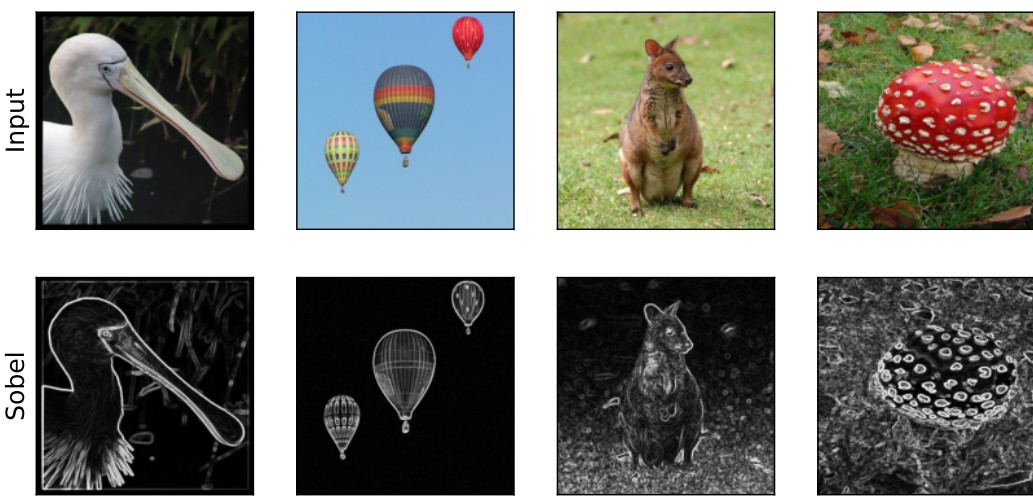

Figure S3: Visual examples of the shape images using Sobel operator

## C   CCL ALGORITHM

---

**Algorithm 2** Cognitive Continual Learner (CCL)

---

**Input:** Dataset $\mathcal{D}_t$, Buffer $\mathcal{B}$
**Initialize:** Three networks: Encoder and classifier $f$ parameterized by $\theta_{WM}$, $\theta_{SM}$, and $\theta_{IBL}$
1: **for** all tasks $t \in 1, 2...T$ **do**
2:     Sample mini-batch: $(x_c, y_c) \sim \mathcal{D}_t$
3:     Extract shape images: $x_{c_s} = \mathbb{B}(x_c)$ where $\mathbb{B}$ is a Sobel filter
4:     $\mathcal{L}_{Sup_{WM}} = \mathcal{L}_{CE}(f(x_c; \theta_{WM}), y_c)$
5:     $\mathcal{L}_{Sup_{IBL}} = \mathcal{L}_{CE}(f(x_{c_s}; \theta_{IBL}), y_c)$
6:     **if** $\mathcal{B} \neq \emptyset$ **then**
7:         Sample mini-batch: $(x_b, y_b) \sim \mathcal{B}$
8:         Extract shape images: $x_{b_s} = \mathbb{B}(x_b)$
9:         Calculate the supervised loss:
10:         $\mathcal{L}_{Sup_{WM}} \mathrel{+}= \mathcal{L}_{CE}(f(x_b; \theta_{WM}), y_b)$
11:         $\mathcal{L}_{Sup_{IBL}} \mathrel{+}= \mathcal{L}_{CE}(f(x_{b_s}; \theta_{IBL}), y_b)$
12:         Knowledge sharing from semantic memory to working model and inductive bias learner:
13:         $\mathcal{L}_{KS_{WM}} = \mathbb{E}\|f(x_b; \theta_{SM}) - f(x_b; \theta_{WM})\|_2^2$
14:         $\mathcal{L}_{KS_{IBL}} = \mathbb{E}\|f(x_b; \theta_{SM}) - f(x_{b_s}; \theta_{IBL})\|_2^2$
15:     Bidirectional knowledge sharing between working model and inductive bias learner:
16:     $\mathcal{L}_{biKS} = \underset{x \sim D_t \cup \mathcal{B}}{\mathbb{E}} \|f(x; \theta_{WM}) - f(x_s; \theta_{IBL})\|_2^2$
17:     Calculate total loss:
18:     $\mathcal{L}_{WM} = \mathcal{L}_{Sup_{WM}} + \lambda \mathcal{L}_{biKS} + \lambda' \mathcal{L}_{KS_{WM}}$
19:     $\mathcal{L}_{IBL} = \mathcal{L}_{Sup_{IBL}} + \gamma \mathcal{L}_{biKS} + \gamma' \mathcal{L}_{KS_{IBL}}$
20:     Update both working model and inductive bias learner: $\theta_{WM}, \theta_{IBL}$
21:     Stochastically update semantic memory:
22:     Sample $s \sim U(0, 1)$;
23:     **if** $s < r$ **then**
24:         $\theta_{SM} = d \cdot \theta_{SM} + (1 - d) \cdot \theta_{WM}$
25:     Update memory buffer $\mathcal{B}$
    **return** model $\theta_{SM}$

---

# D  SETTING AND DATASETS

We evaluate all methods in different CL settings. Van de Ven & Tolias (2019) describes three different settings based on increasing difficulty: task incremental learning (Task-IL), domain incremental learning (Domain-IL), and class incremental learning (Class-IL). In Class-IL, each new task consists of novel classes, and the network must learn both new classes while retaining information about the old ones. Task-IL is similar to Class-IL but assumes that the task labels are accessible at both training and inference. In Domain-IL, the classes remain the same for each task, but the distribution varies for each task. We report the results for all three settings on the relevant datasets. CLass-IL is relatively the most complex setting of the three and is widely studied; however, there are some assumptions that simplify this setting to be realistic. Mi et al. (2020) highlighted some of the limitations of Class-IL, such as the assumption of the same number of classes across different tasks, no reappearance of classes, and the sample distribution per class is well balanced. Hence, Generalized Class-IL (GCIL) was suggested to overcome these limitations and introduce a more realistic setting. GCIL is a more generalized CL setting, where the number of classes in each task is not fixed, and the classes can reappear with varying sample sizes. GCIL samples the number of classes and samples from a probabilistic distribution. The two variations are Uniform (fixed uniform sample distribution over all classes) and Longtail (with class imbalance).

We report results on all three settings: Task-IL, Domain-IL, and CLass-IL. Furthermore, we also consider the GCIL setting for one of the dataset as an additional evaluation setting. All reported results are averaged over three random seeds.

# E  HYPERPARAMETERS

We utilize a small validation set to tune the hyperparameters for all methods. For Seq-CIFAR10, we report the results of the original articles (Buzzega et al., 2020; Cha et al., 2021; Caccia et al., 2021; Arani et al., 2021). For the other datasets, we ran a grid search over the hyperparameters reported in the paper for a similar dataset. For Seq-CIFAR100 and GCIL-CIFAR100, we formed the search range using the Seq-CIFAR10 huperparameters as a reference point. Search ranges are shown in Table S3.

Domain$^2$L dataset is more complex compared to the CIFAR versions and includes images of larger sizes. Hence, we consider the Seq-TinyImagenet hyperparameters in the respective paper as the reference point for further tuning. The learning rate $lr$, the number of epochs, and the batch size are similar across the datasets. The ema update rate $r$ is lower for more complex datasets, as shown in CLS-ER. $r$ is chosen in the range of $[0.01, 0.1]$ with a step size of $0.02$ for CLS-ER and CCL. The different hyperparameters chosen for the baselines, after tuning, are shown in Table S4.

The different hyperparameters chosen for CCL are shown in Table S5. The parameters : $lr$, batch size, number of epochs are uniform across all datasets. The stochastic update rate and decay parameter are similar to CLS-ER. The hyperparaneters and stable across settings and datasets and and also compliment each other. The loss balancing weights are reported as four different parameters for clarity, however, they show similar pattern. Therefore, CCL does not require extensive fine-tuning across different datasets and settings.

# F  COMPLEXITY

We discuss the computational complexity aspect of our proposed method. CCL involves three networks during training; however, in inference, only a single network is used (SM module). Therefore, for inference purposes, the MAC count, the number of parameters, and computational capacity remain the same as the other single-network methods.

The training cost requires three forward passes, as it consists of three different modules. ER, DER++, CO$^2$L and ER-ACE have a single network. CLS-ER also has three networks and therefore requires 3 forward passes. CCL has training complexity similar to CLS-ER; however, it outperforms CLS-ER in all provided metrics.

Table S3: Search ranges for tuning hyperparameters

| Method | Hyperparameters | Search Range |
|--------|----------------|--------------|
| ER | $lr$ | [0.01, 0.03, 0.1, 0.5] |
| DER++ | $lr$ | [0.01, 0.03, 0.1] |
| | $\alpha$ | [0.1, 0.2, 0.5] |
| | $\beta$ | [0.5, 1.0] |
| $CO^2L$ | $lr$ | [0.01, 0.03, 0.1] |
| | $\tau$ | [0.01, 0.1, 0.5] |
| | $k$ | [0.2, 0.5] |
| | $k^*$ | [0.01, 0.05] |
| | $e$ | [100, 150] |
| ER-ACE | $lr$ | [0.01, 0.03, 0.1, 0.5] |
| CLS-ER | $lr$ | [0.01, 0.03, 0.1] |
| | $\lambda$ | [0.1, 0.2, 0.3] |
| | $r_p$ | [0:1:0.1] |
| | $r_s$ | [0:1:0.1] |
| | $\alpha_p$ | [0.99, 0.999] |
| | $\alpha_s$ | [0.99, 0.999] |
| CCL | $lr$ | [0.01, 0.03, 0.1] |
| | $r$ | [0:1:0.1] |
| | $d$ | [0.99, 0.999] |
| | $\lambda$ | [0.01, 0.1] |
| | $\gamma$ | [0.01, 0.1] |
| | $\lambda^,$ | [0.01, 0.1] |
| | $\gamma^,$ | [0.01, 0.1] |

On the memory front, similar to all methods, we save memory samples based on the memory budget allotted (200 and 500 in the experiments). There are no additional memory requirements, as we do not save any extra information (such as logits) to be used later in our objectives.

## G    OTHER METRICS

Forward transfer, backward transfer, and forgetting are other metrics (Lopez-Paz & Ranzato, 2017) used in CL literature. These metrics are estimated from the model checkpoint after a task is completed, as this checkpoint has the highest accuracy for that particular task. However, this does not hold true for our method, which utilizes the stochastically updated model for inference and evaluation purposes. The SM module assimilates knowledge from the working model and is updated stochastically by the exponential moving average. It achieves highest accuracy on previous tasks while also learning the new tasks. Therefore, the results may be misleading.

However, we evaluate backward transfer (-1×forgetting) by considering the best accuracy of the SM module after a particular task and then finding the difference between this and the final accuracy. Taking into account Figure S2, if we use the backward metric formula directly, we get a positive backward transfer for CCL in Tasks 3 and 4 (due to the SM model achieving high accuracy in the previous task while also learning the new task at that stochastic update frequency); therefore, we pick the maximum one and subtract it from the last row. We report the values for the metrics in Table S7. CCL fares better in backward transfer (or forgetting) compared to other techniques.

## H    EXTENDED RELATED WORKS

One of the modules (IBL) in CCL utilizes the inductive bias in terms of shape to produce more generic representations. There are several works Geirhos et al. (2018) that showcase the texture bias problem of neural networks. Several techniques have been introduced to reduce texture bias

Table S4: Selected hyperparameters for all baselines.

| Dataset | $|\mathcal{B}|$ | Method | Hyperparameters |
|---|---|---|---|
| Seq-CIFAR100 | 200 | ER | $lr$=0.1 |
| | | DER++ | $lr$=0.03, $\alpha$=0.1, $\beta$=0.5 |
| | | CO$^2$L | $lr$:0.5, $\tau$:0.5, $\kappa$:0.2, $\kappa^*$:0.01, $e$:100 |
| | | ER-ACE | $lr$=0.01 |
| | | CLS-ER | $lr$=0.1 $\lambda$=0.15, $r_p$=0.1, $r_s$=0.05, $\alpha_p$=0.999, $\alpha_s$=0.999 |
| | 500 | ER | $lr$=0.1 |
| | | DER++ | $lr$=0.03, $\alpha$=0.1, $\beta$=0.5 |
| | | CO$^2$L | $lr$:0.5, $\tau$:0.5, $\kappa$:0.2, $\kappa^*$:0.01, $e$:100 |
| | | ER-ACE | $lr$=0.01 |
| | | CLS-ER | $lr$=0.1 $\lambda$=0.15, $r_p$=0.1, $r_s$=0.05, $\alpha_p$=0.999, $\alpha_s$=0.999 |
| GCIL-CIFAR100 | 200 | ER | $lr$=0.1 |
| | | DER++ | $lr$=0.03, $\alpha$=0.5, $\beta$=0.1 |
| | | CO$^2$L | - |
| | | ER-ACE | $lr$=0.1 |
| | | CLS-ER | $lr$=0.1 $\lambda$=0.1, $r_p$=0.7, $r_s$=0.6, $\alpha_p$=0.999, $\alpha_s$=0.999 |
| | 500 | ER | $lr$=0.1 |
| | | DER++ | $lr$=0.03, $\alpha$=0.2, $\beta$=0.1 |
| | | CO$^2$L | - |
| | | ER-ACE | $lr$=0.1 |
| | | CLS-ER | $lr$=0.1 $\lambda$=0.1, $r_p$=0.7, $r_s$=0.6, $\alpha_p$=0.999, $\alpha_s$=0.999 |
| Domain$^2$IL | 200 | ER | $lr$=0.1 |
| | | DER++ | $lr$=0.03, $\alpha$=0.1, $\beta$=1.0 |
| | | CLS-ER | $lr$=0.05 $\lambda$=0.1, $r_p$=0.08, $r_s$=0.04, $\alpha_p$=0.999, $\alpha_s$=0.999 |
| | 500 | ER | $lr$=0.1 |
| | | DER++ | $lr$=0.03, $\alpha$=0.5, $\beta$=0.1 |
| | | CLS-ER | $lr$=0.05 $\lambda$=0.1, $r_p$=0.08, $r_s$=0.05, $\alpha_p$=0.999, $\alpha_s$=0.999 |

Table S5: Selected hyperparameters for CCL across different settings.

| | $|\mathcal{B}|$ | lr | batch size | #epochs | $r$ | $d$ | $\lambda$ | $\gamma$ | $\lambda'$ | $\gamma'$ |
|---|---|---|---|---|---|---|---|---|---|---|
| Seq-CIFAR10 | 200 | 0.03 | 32 | 50 | 0.2 | 0.999 | 0.1 | 0.1 | 0.1 | 0.1 |
| | 500 | 0.03 | 32 | 50 | 0.2 | 0.999 | 0.1 | 0.1 | 0.1 | 0.1 |
| Seq-CIFAR100 | 200 | 0.03 | 32 | 50 | 0.06 | 0.999 | 0.1 | 0.01 | 0.1 | 0.01 |
| | 500 | 0.03 | 32 | 50 | 0.08 | 0.999 | 0.1 | 0.01 | 0.1 | 0.01 |
| GCIL-CIFAR100 | 200 | 0.03 | 32 | 50 | 0.09 | 0.999 | 0.1 | 0.01 | 0.1 | 0.01 |
| | 500 | 0.03 | 32 | 50 | 0.2 | 0.999 | 0.1 | 0.01 | 0.1 | 0.01 |
| DN4IL | 200 | 0.03 | 32 | 50 | 0.06 | 0.999 | 0.1 | 0.01 | 0.1 | 0.01 |
| | 500 | 0.03 | 32 | 50 | 0.08 | 0.999 | 0.1 | 0.01 | 0.1 | 0.01 |

Table S6: Fifteen different natural corruptions

| | |
|---|---|
| Corruptions | Gaussian Noise, Impulse Noise, Shot noise, Speckle noise |
| | Defocus blur, Glass blur, Motion blur, Zoom blur, Gaussian blur |
| | Brightness, Contrast, Fog, Frost, Snow |
| | Elastic Transformation, JPEG compression, Pixelate, Spatter, Saturate |

and improve representations. Geirhos et al. (2018) increases shape bias by adding multiple stylized images along with the original images used for training. Styles of artistic paintings are transferred

Table S7: Backward transfer metric on Seq-CIFAR10 dataset

| Method | Backward Transfer |
|--------|-------------------|
| ER | -61.75 |
| DER++ | -33.45 |
| CLS-ER | -23.47 |
| CCL | **-6.07** |

Table S8: Accuracy on the proposed *DN4IL* dataset for the Domain-IL setting. CCL shows a significant improvement in all disparate and challenging domains.

| $\mathcal{B}$ | Method | real | clipart | infograph | painting | sketch | quickdraw | Acc |
|---|---|---|---|---|---|---|---|---|
| - | JOINT | | | | | | | $59.93_{\pm1.07}$ |
| | SGD | $9.98_{\pm0.54}$ | $19.97_{\pm0.31}$ | $2.32_{\pm0.20}$ | $6.58_{\pm0.34}$ | $14.91_{\pm0.04}$ | $71.23_{\pm0.17}$ | $20.83_{\pm0.24}$ |
| 200 | ER | $20.08_{\pm0.45}$ | $26.37_{\pm0.35}$ | $5.56_{\pm0.39}$ | $13.92_{\pm0.91}$ | $23.69_{\pm1.54}$ | $69.95_{\pm0.56}$ | $26.59_{\pm0.31}$ |
| | DER++ | $33.66_{\pm1.65}$ | $37.24_{\pm0.64}$ | $9.80_{\pm0.63}$ | $24.16_{\pm1.17}$ | $34.37_{\pm2.00}$ | $69.26_{\pm0.79}$ | $34.75_{\pm0.87}$ |
| | CLS-ER | $45.53_{\pm0.88}$ | $49.17_{\pm1.12}$ | $15.79_{\pm0.48}$ | $35.80_{\pm0.64}$ | $48.03_{\pm0.85}$ | $54.40_{\pm1.25}$ | $40.83_{\pm1.07}$ |
| | CCL | $47.52_{\pm0.25}$ | $54.69_{\pm0.10}$ | $15.70_{\pm0.33}$ | $37.54_{\pm0.30}$ | $51.98_{\pm0.96}$ | $58.80_{\pm0.18}$ | $\mathbf{44.23_{\pm0.05}}$ |
| 500 | ER | $27.54_{\pm0.05}$ | $31.89_{\pm0.93}$ | $7.89_{\pm0.45}$ | $19.39_{\pm1.02}$ | $28.36_{\pm1.35}$ | $70.96_{\pm0.10}$ | $31.01_{\pm0.62}$ |
| | DER++ | $44.49_{\pm1.39}$ | $46.17_{\pm0.35}$ | $14.01_{\pm0.23}$ | $33.44_{\pm0.90}$ | $43.59_{\pm1.11}$ | $69.53_{\pm0.29}$ | $41.87_{\pm0.63}$ |
| | CLS-ER | $49.85_{\pm0.88}$ | $51.41_{\pm0.34}$ | $18.17_{\pm0.08}$ | $37.94_{\pm0.94}$ | $49.02_{\pm1.57}$ | $55.63_{\pm0.71}$ | $43.41_{\pm0.80}$ |
| | CCL | $54.77_{\pm0.15}$ | $60.37_{\pm0.75}$ | $19.35_{\pm0.39}$ | $44.50_{\pm0.43}$ | $56.34_{\pm0.53}$ | $60.61_{\pm1.73}$ | $\mathbf{49.32_{\pm0.23}}$ |

to Imagenet dataset to create stylized-imagenet. Style-transfer is performed using adaptive instance stylization (ADaIN) and is an additional offline process. Chen et al. (2016) uses generative techniques (InfoGan) to synthesize images that are less biased to texture. Li et al. (2020) also creates an augmented dataset by blending the texture of one image and shape of another image in the training set to create a new image. All of these techniques, then merge both the original and synthesized data to train the network on a bigger dataset. However, training a single network with these different distributions leads to learning sub-optimal representations. This is also shown in the results in Geirhos et al. (2018), where they had to do an additional fine-tuning on the original dataset to achieve better results on the original data. Also, synthesizing and generating new data is expensive and might come with an unaccounted bias.

CCL on the other hand, tries to leverage on the under-utilized implicit shape information with minimal overhead. There is no requirement of additional data, generative networks and the RGB and shape data are not combined together to make one big training dataset. The RGB image is fed to one network and the shape information is learnt by another network (IBL) and the supervision is provided via a knowledge transfer between these two networks and is mutual. Each network has enough flexibility to learn on its own feature while also getting guidance from the other feature.

## I    *DN4IL*

We introduce a new dataset for the Domain-IL setting. It is a subset of the standard DomainNet dataset (Peng et al., 2019) used in domain adaptation. It consists of six different domains - real, clipart, infograph, painting, quickdraw, and sketch. The shift in distribution between domains is challenging. Few examples can be seen in Figure S4.

Each domain includes 345 classes, and the overall dataset consists of $\sim$59000 samples. The classes have redundancy, and also evaluating on the whole dataset can be computationally expensive for CL settings. Therefore, we create a subset by grouping semantically similar classes into 20 super categories (considering class overlap between other standard datasets can also facilitate OOD analysis). Each super category has five classes each, which results in a total of 100 classes. The specifications of the classes are given in Table S9. The dataset consists of 67080 training images and 19464 test images. The image size for all experiments is chosen as 64×64 (the normalize transform is not applied in the augmentations).

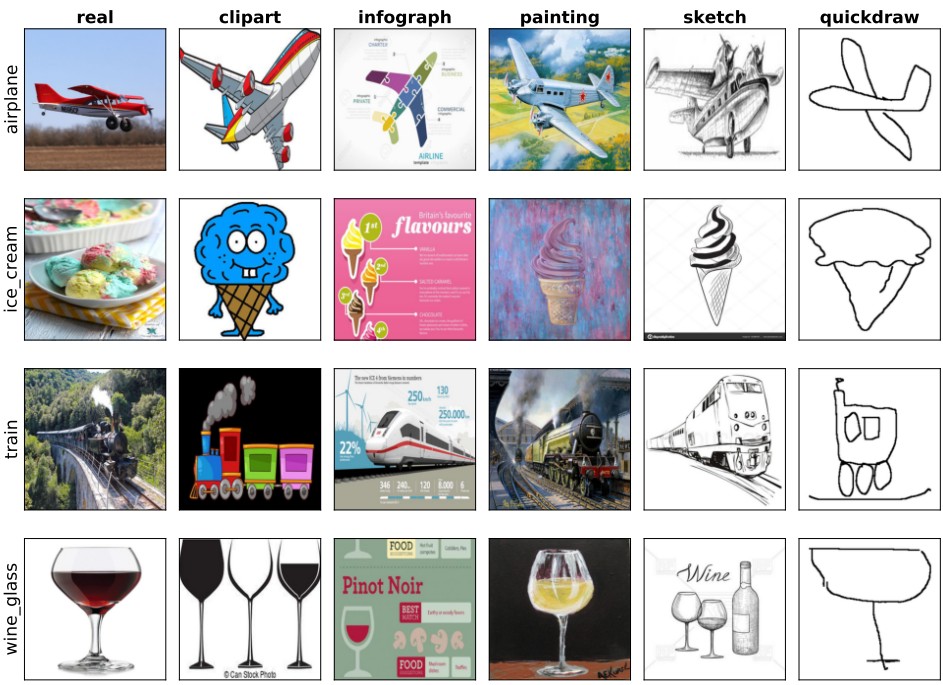

Figure S4: Visual examples of *DN4IL* dataset

Table S9: Details on supercategory and classes in *DN4IL* dataset.

|    | supercategory | class | | | | |
|----|---------------|-------|---|---|---|---|
| 1  | small animals | mouse | squirrel | rabbit | dog | raccoon |
| 2  | medium animals | tiger | bear | lion | panda | zebra |
| 3  | large animals | camel | horse | kangaroo | elephant | cow |
| 4  | aquatic mammals | whale | shark | fish | dolphin | octopus |
| 5  | non-insect invertebrates | snail | scorpion | spider | lobster | crab |
| 6  | insects | bee | butterfly | mosquito | bird | bat |
| 7  | vehicle | bus | bicycle | motorbike | train | pickup_truck |
| 8  | sky-vehicle | airplane | flying_saucer | aircraft_carrier | helicopter | hot_air_balloon |
| 9  | fruits | strawberry | banana | pear | apple | watermelon |
| 10 | vegetables | carrot | asparagus | mushroom | onion | broccoli |
| 11 | music | trombone | violin | cello | guitar | clarinet |
| 12 | furniture | chair | dresser | table | couch | bed |
| 13 | household electrical devices | clock | floor_lamp | telephone | television | keyboard |
| 14 | tools | saw | axe | hammer | screwdriver | scissors |
| 15 | clothes & accessories | bowtie | pants | jacket | sock | shorts |
| 16 | man-made outdoor | skyscraper | windmill | house | castle | bridge |
| 17 | nature | cloud | bush | ocean | river | mountain |
| 18 | food | birthday_cake | hamburger | ice_cream | sandwich | pizza |
| 19 | stationary | calendar | marker | map | eraser | pencil |
| 20 | household items | wine_bottle | cup | teapot | frying_pan | wine_glass |

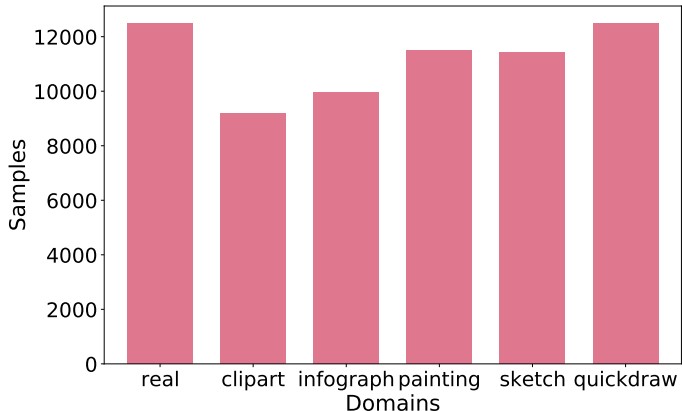

Figure S5: Number of samples per domain in *DN4IL* dataset.

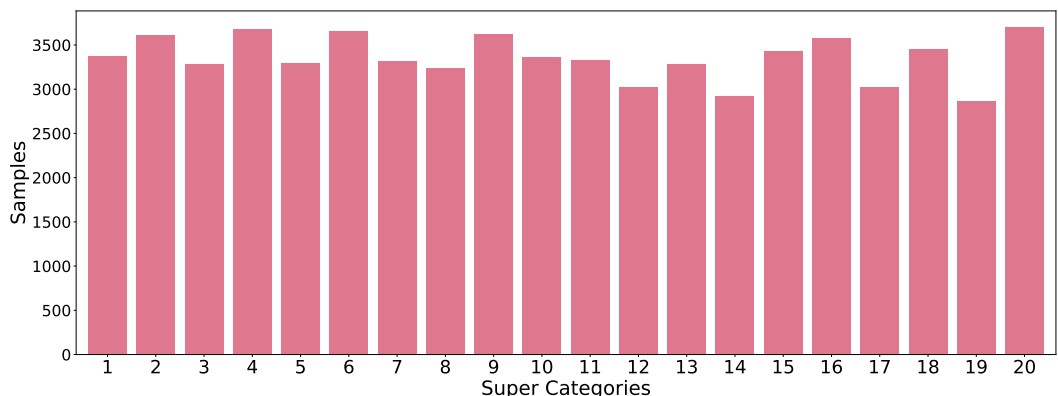

Figure S6: Number of samples per super category in *DN4IL* dataset.

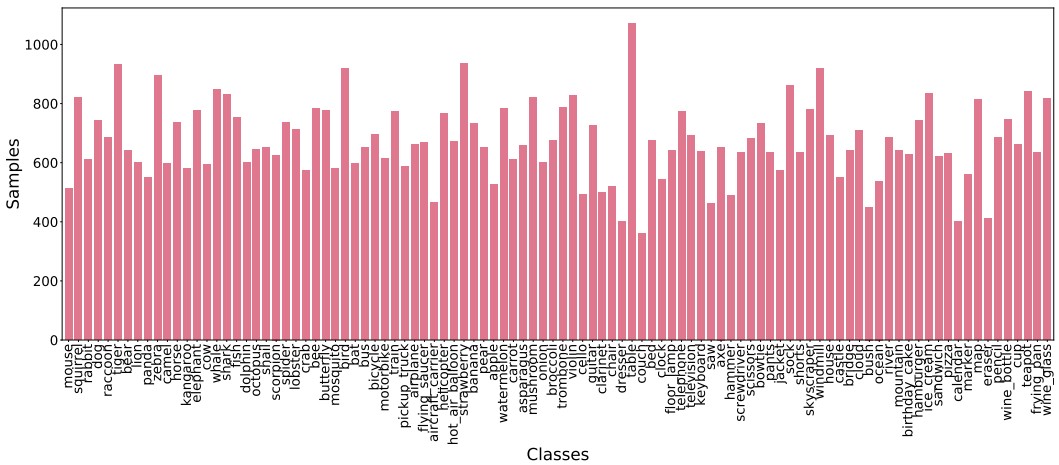

Figure S7: Number of overall samples per class in *DN4IL* dataset.

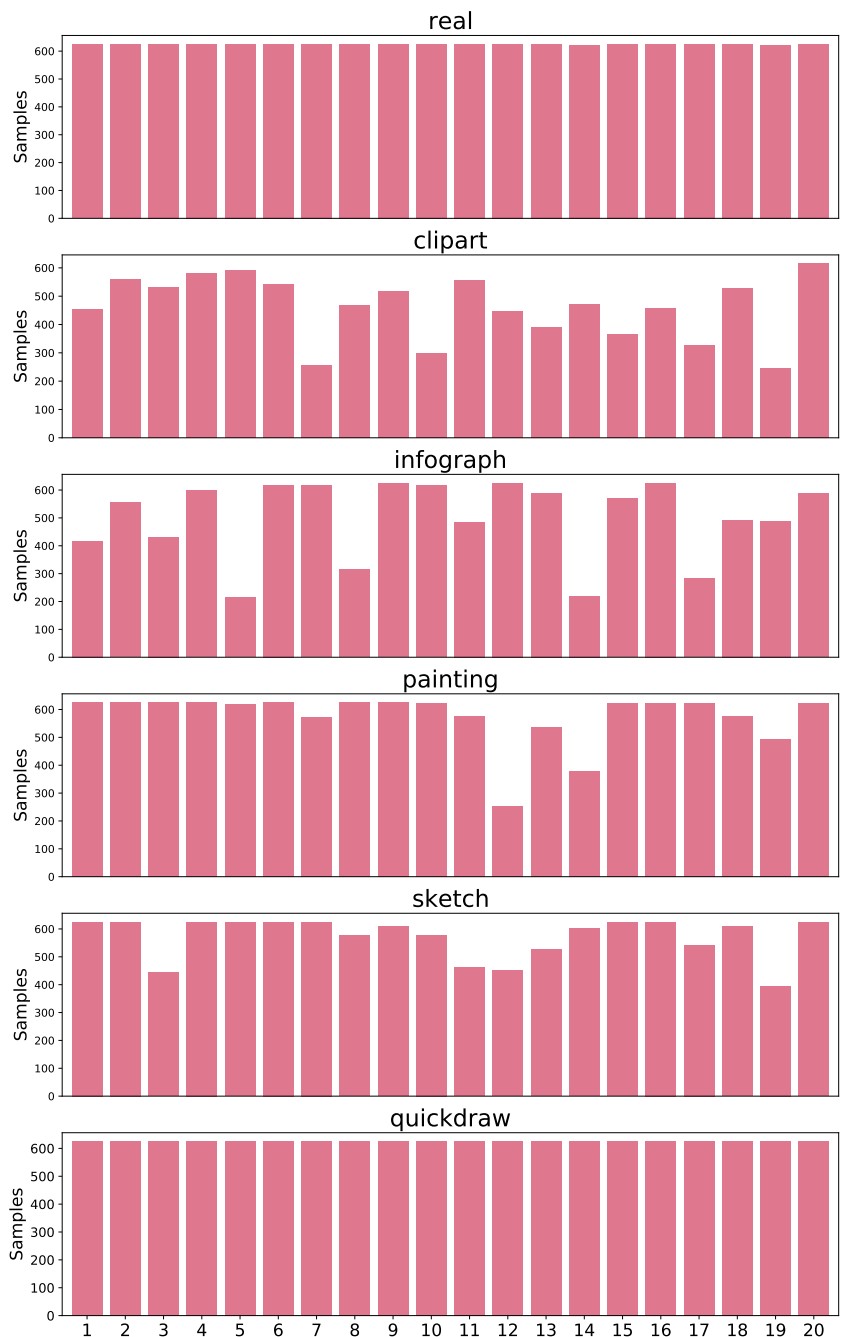

Figure S8: Number of samples per supercategory for each domain in *DN4IL* dataset.

