# OpenReview forum: "A Cognitive-inspired Multi-Module Architecture for Continual Learning"
_ICLR.cc/2023/Conference — Submitted to ICLR 2023_

### Official Review · Reviewer_6gTb · 2022-10-23

**Confidence:** 4
**Correctness:** 3
**Technical Novelty And Significance:** 3
**Empirical Novelty And Significance:** 3
**Recommendation:** 6

**Clarity, Quality, Novelty And Reproducibility:**

The paper is decently written but lacking clarity with respect to the details of the implementation, which makes it impossible to reproduce, given that no code is provided.

**Strength And Weaknesses:**

- The method in the paper has a number of interesting features:
    - (i) the knowledge sharing between multiple modules. Knowledge sharing via distillation between two modules (that usually evolve at different timescales) has been used in continual learning in a number of existing methods (e.g. Progress and Compress, Schwarz et al 2018), but CCL enforces consistency between 3 modules.
    - (ii) enforcing consistency between two different views of the data (the raw input and the shape information)
- The empirical results are strong, and are run on relevant benchmarks comparing against suitable baselines.

Opportunities for improvement / questions
- There are very important details missing about the implementation that make it difficult to fully understand the method, and certainly impossible to reproduce (given that no code is provided).
    - (i) There seems to be no explanation of how the shape information is extracted from the data and represented to the inductive bias learner. This appears to be a key feature to the model. Indeed in 5.3 it is even stated that the primary benefit of CCL comes from the shape information.
    - (ii) The stochastic momentum update does not seem to be described - what is the nature of the stochasticity in this update and how important is it to the performance of the model?
- In the ablations it would be interesting to see what would break the model - all the ablations seem to fair quite well, for instance much better than the experience replay baseline, so it is difficult to understand which aspects of the model are the most important. Maybe some further ablations are needed to ascertain this, e.g. how would the working model perform by itself without either the SM or the IBL? Would it be correct to say that this should be similar to the performance of ER?
- It is mentioned that some of the results in Table 1 are taken from the original papers but some are run again by the authors - but it is not clear which are rerun and which are not.
- The plasticity and stability metrics used are not obviously reflective of the conventional meaning of the terms. Stability is computed as the average accuracy all tasks 1:T-1 after learning of the final task T, but this metric is highly dependent on the initial performance on tasks 1:T-1 - would it not be more informative to report a more standard metric like forgetting or backward transfer (Lopez-Paz et al. 2017)? Also, why do the plasticity and stability values in Figure 2 go above 100%?
- What is the rationale for using an MSE loss for the Knowledge-sharing loss rather than say a Kullback-Leibler loss, when the outputs of the WM and IBL networks are probability vectors?

**Summary Of The Paper:**

This paper presents a method for continual learning (Cognitive Continual Learner, CCL) with three components: a ‘working model’ that learns to classify from raw input, an ‘inductive bias learner’ that learns to classify from a processed version of the input in the form of shape information, and a ’semantic learner’, which is a stochastic moving average of the parameters of the working model. Apart from the classification losses on the first two modules, there are also few auxiliary ‘knowledge sharing’ objectives: a bidirectional MSE loss between the outputs of the working model and the inductive bias learner, enforcing them to make consistent predictions, and MSE losses between the output of the semantic learner and each of the working model and inductive bias learner, constraining each of them to not veer too far from historical predictions. The model also uses a replay buffer maintained with reservoir sampling. Experiments in both the task incremental and class incremental settings are run on Sequential CIFAR-10, Sequential CIFAR-100, GCIL-CIFAR100 (which blurs boundaries between tasks) and a custom dataset introduced in the paper called Domain2IL, which is a subset of the existing DomainNet dataset, curated to make the class distributions more balanced and uniformly distributed. CCL is compared to a number of replay-based baselines and is shown to be the best performing across all tasks. Some additional analyses are run to demonstrate the robustness of the method, how it fairs with the plasticity-stability tradeoff, and some ablations of the components of the model.

**Summary Of The Review:**

Update: I have increased my score to 6 after authors responded to my concerns.
------
This paper presents an interesting method for continual learning, that features enforcing consistency between multiple modules, that is justified by strong experimental results on a number of CL visual classification benchmarks comparing to a number of relevant replay-based baselines. In its current form, the paper is lacking a number of important implementational details that render it entirely unreproducible (described in more detail above), which is the main reason I think it cannot be accepted as is. It is also difficult to tell from the ablations which components of model architecture and loss functions are most important to performance, as all the ablations do well - I think further ablations are needed in order to determine this (as descibed above), which would provide a lot more insight into why the model works well. I would be willing to increase my score if these two points were addressed.

---

> ### Author Response · Authors · 2022-11-13
> **Reply to Reviewer 6gTb**
>
> We would like to thanks the reviewer for the feedback and suggestions and we are glad that you found knowledge sharing between 3 modules interesting.
>
> >  There seems to be no explanation of how the shape information is extracted from the data and represented to the inductive bias learner. This appears to be a key feature to the model. Indeed in 5.3 it is even stated that the primary benefit of CCL comes from the shape information.
>
>  Thanks for bringing this to our attention. Shape extraction is crucial and we have added the detailed algorithm along with some visual examples in Section B in the Appendix. We use the Sobel filter to extract shape. We considered multiple algorithms and finalized sobel based on the precision, smoothness and quality of the edges and the overall shape.
>
> > The stochastic momentum update does not seem to be described - what is the nature of the stochasticity in this update and how important is it to the performance of the model?
>
> Sorry for lack of information on this crucial topic. The semantic memory module assimilates information and utilizes exponential moving average of the working module. As our method does not rely on the task boundary, we cannot update the model at the task boundary. Fixing a “n” and updating the module every nth iteration is not an optimal approach as it offers no variation and can create a bias. Hence, to cater for general IL settings, we update it stochastically. Further, the stochastic update empirically performs better. To ensure lucidity, we updated the equations and also the algorithm.
>
> > In the ablations it would be interesting to see what would break the model - all the ablations seem to fair quite well, for instance much better than the experience replay baseline, so it is difficult to understand which aspects of the model are the most important. Maybe some further ablations are needed to ascertain this, e.g. how would the working model perform by itself without either the SM or the IBL? Would it be correct to say that this should be similar to the performance of ER?
>
> Yes, it is correct that the model without SM or IBL would be similar to ER. We have updated Table 2 (also shown below) in the paper with an additional row.  We have also simplified the table for better clarity and added some explanation. As the goal was to incorporate multiple components and check the gain in performance, we considered these combinations and to check the generality, we performed it on two different datasets. Each individual component is quite stable and helps in improving over the ER,  and further there are no convergence or exploding problems that can result in breaking of the model. Each component improves the performance and the maximum gain is obtained when all the components are combined together. In the table, ER for Domain dataset (without any of our components) is 26.59 while each component keeps enhancing the results. The highest accuracy is obtained when all components are combined.
>
> | SM  | IBL | KS (WM-IBL) | Seq-CIFAR10 | Domain2IL |
> |-----|-----|-------------|-------------|------------|
> |  ✔   |   ✔  |    ✔         | 70.04±1.07  | 44.23±0.05 |
> |  ✔   |  ✔   |    ✗         | 69.28±1.34  | 40.35±0.34 |
> |  ✔   |  ✗   |         -    | 69.21±1.46  | 39.76±0.56 |
> |  ✗   |  ✔   |      ✔       | 64.61±1.22  | 37.33±0.01 |
> |  ✗   |  ✗   |     ✗        | 44.79±1.86  | 26.59±0.3  |
>
> If we could provide further ablations, please let us know what extra combinations we can report?
>
> > It is mentioned that some of the results in Table 1 are taken from the original papers but some are run again by the authors - but it is not clear which are rerun and which are not.
>
> We apologize for the lack of clarity on this topic. We have added a detailed section to address all these concerns (Section E).  Seq-CIFAR10 for all methods were reported from the papers, while for CIFAR-100, GCIL-CIFAR100 and Domain2IL, we ran all the methods. The procedure of how we selected the parameter grid for fine-tuning baseline and also our method is provided with tables and explanation in Section E. Please let us know if it is more clear now.

---

> > ### Author Response · Authors · 2022-11-13
> > **Reply to Reviewer 6gTb (2/2)**
> >
> > > The plasticity and stability metrics used are not obviously reflective of the conventional meaning of the terms. Stability is computed as the average accuracy all tasks 1:T-1 after learning of the final task T, but this metric is highly dependent on the initial performance on tasks 1:T-1 - would it not be more informative to report a more standard metric like forgetting or backward transfer (Lopez-Paz et al. 2017)? Also, why do the plasticity and stability values in Figure 2 go above 100%?
> >
> > Firstly, thank you for notifying the mistake in Figure 2. We took the absolute values, without averaging, while plotting the graph. This has been rectified now.
> >
> > We report the average performance across tasks which also encompass the plasticity-stability information. To supplement this with a more specific metric that calculates the plasticity-stability trade-off, we used this metric that is defined in [5].
> > However, this feedback is a good point and it does depend on the initial performance and might not be completely accurate in all the cases.
> >
> > Regarding other metrics such as forward and backward knowledge transfer, we had intended to include them, but we ran into a quandary over it. These metrics are estimated from the model checkpoint, after a task is completed, as this checkpoint has the highest accuracy for that particular task. However, this does not hold true to our technique, which utilizes the stochastically updated model or the Semantic memory model for inference and evaluation purposes. The SM module assimilates knowledge from the working model and is updated at different frequencies by the exponential moving average. It achieves highest accuracy on previous tasks while also learning the new tasks. Owing to these reasons, the evaluation of these metrics for CCL might not be consistent and can be misleading while compared with techniques.  For example, considering Figure S2, if we use the backward metric formula directly, we get a positive backward transfer for CCL in tasks 3 and 4 (due to the SM model achieving high accuracy on previous tasks while also learning new tasks at that stochastic update frequency).
> >
> > However, one way for backward transfer, could be to find the best accuracy of the SM module on a particular task and then finding the difference between this and the final accuracy to find the backward transfer. We have explained this and reported preliminary results on one dataset in Section G in appendix. CCL evidently has better backwards transfer.  We can report on other datasets as well and complete the table.
> >
> > | Method | Backward  Transfer |
> > |:------:|:------------------:|
> > |   ER   |       -61.75       |
> > |  DER++ |       -33.45       |
> > | CLS-ER |       -23.47       |
> > |   CCL  |        -6.07       |
> >
> > > What is the rationale for using an MSE loss for the Knowledge-sharing loss rather than say a Kullback-Leibler loss, when the outputs of the WM and IBL networks are probability vectors?
> >
> > Many recent works on Knowledge distillation (KD) are resorting to MSE loss, compared to KL divergence. Some works [1] rationalize that the softmax function acts as a squashing function that results in information loss in the probability. Hence matching logits using MSE loss is being used in latest works involving KD [2][3]. [4] also performs evaluation using different consistency loss objectives and shows the superior performance of MSE in CL settings. Further, MSE also offers a stricter condition, compared to the flexible KL divergence, which is useful when allowing for effective knowledge transfer between different distributions (such as RGB, shape), such as our method.
> >
> > [1] Liu, Xuan, et al. "Improving the interpretability of deep neural networks with knowledge distillation." 2018 IEEE International Conference on Data Mining Workshops (ICDMW). IEEE, 2018. \
> > [2] Kang, et al. "Class-Incremental Learning by Knowledge Distillation with Adaptive Feature Consolidation." CVPR 2022. \
> > [3] Buzzega, et al. “Dark experience for general continual learning: a strong, simple baseline.” NeurIPS 2020. \
> > [4] Bhat, et al "Consistency is the key to further mitigating catastrophic forgetting in continual learning." CoLLAs (2022). \
> > [5] Sarfraz, et al., "SYNERgy between SYNaptic Consolidation and Experience Replay for General Continual Learning." CoLLAs (2022).
> >
> >
> > Please let us know if the concerns regarding the reproducibility and ablations are addressed. Also, we were planning to publish the code online after acceptance.\
> > We would like to engage more and discuss and provide more details if necessary.

---

> ### Author Response · Authors · 2022-11-18
> **General Response to 6gTb**
>
> As the first stage of discussion is nearing the deadline, we would like to take the opportunity to check if we have addressed all the concerns and questions. We would be happy to engage more and provide more details if necessary. Thank you.

---

### Official Review · Reviewer_BtBz · 2022-10-24

**Confidence:** 5
**Correctness:** 4
**Technical Novelty And Significance:** 2
**Empirical Novelty And Significance:** 3
**Recommendation:** 6

**Clarity, Quality, Novelty And Reproducibility:**

The paper is clear but no clear motivation behind the different used losses.  I am mostly worries about  the hyper-parameters. It seems they were picked based on overall performance. Usually, hyper-parameters are set based on held-out set, e.g., a subset of the sequence  or set for one sequence and fixed for others. Here there is a large number of hyper-parameters with wide range and no discussion on how they were set.

**Details Of Ethics Concerns:**

A solution to continual learning is proposed that doesn't seem to flag any ethical concern.

**Strength And Weaknesses:**

+ New brain inspired solution to continual learning.
+ Good results.
- A part from the cognitive emulation, there doesn't seem good reasons behind the chosen design strategies.
- 3 networks are used increasing the computational footprint of the proposed  solution. Computational aspect is not discussed. A  fair comparison with other methods would be by increasing the buffer size to be equal to the amount of storage due to the extra networks.
- The method requires large number of hyperparameters.
- Not sure why the new domain incremental sequence is introduced. Core50 domain incremental learning sequence could be used.
- Only cifar based sequences are considered, what about ImageNet, TinyImageNet or Core50?
- Not sure why the authors picked only 5 tasks sequence, usually evaluation is done with different number of classes per task in order to show the robustness of the method.
- Not sure that the method is state of the art, there seem to be new published works that show better performance, e.g., PoDNet[1]
- The use of the shape limits the applicability of the method on recognizing objects with discriminating characteristics other than shape, e.g., a horse and a zerbra.
[1] Ashok, Arjun, K. J. Joseph, and Vineeth Balasubramanian. "Class-Incremental Learning with Cross-Space Clustering and Controlled Transfer." ECCV 2022.

**Summary Of The Paper:**

The paper proposes a new continual learning method inspired by human abilities to continually learn with no catastrophic forgetting. Based on aspects in cognitive architecture, 3 modules are combines in the proposed solution.  The authors suggest that the shape of objects is not well leveraged in current CNNs and use a specific module for learning from shapes. The third module acts as a memory module and is updated via momentum updates. Experiments on cifar10 and cifar100 show improvements over compared methods. Also a domain incremental sequence is proposed based on DomainNet benchmark.


**Summary Of The Review:**

A  new cognitive inspired solution to continual learning is proposed with multiple networks. Some aspects are missing and comparisons are limited. To this reviewer, the paper int its current shape is not robust enough.

---

> ### Author Response · Authors · 2022-11-13
> **Reply to Reviewer BtBz (1/3)**
>
> We would like to thank the reviewer for their detailed feedback. We would like to address each of them below.
>
> > A part from the cognitive emulation, there doesn't seem good reasons behind the chosen design strategies.
>
> Thanks for the feedback, we would like to elucidate more on our motivation and reasoning for this work. Cognitive emulation was the main goal in devising the new design strategy. Neuroscience was one of the key inspirations in the origin of neural networks. Human brain is still the most effective intelligent agent that can perform many tasks at ease, that the networks still struggle with (continual learning, transfer learning, robustness, sensitivity to perturbations). Many methods get proposed frequently with minimal changes, which tends to create a trend of research bias of only moving towards nominal increments. The intersection of neuroscience and AI is an exciting field that aims to incorporate knowledge about the workings of the brain, to neural networks, to improve performance at the core working levels. Multiple scientists and pioneers are publishing  neuroscience-inspired works [1][2]. Many scientists (Ex. Yoshua Benjio) are vocal and promoting developing works “that are inspired from the cognitive neuroscience of conscious processing (in particular the global workspace theory and descendants) as they are yet to be satisfactorily incorporated in ML, especially in deep learning. They play a key role in improving robustness, out-of-distribution generalization, transfer learning, and systematic generalization.”
>
> Hence, our work is a preliminary attempt to incorporate cognitive-architecture based elements into the CL algorithm to gauge the gain in reducing forgetting. We hope the promising results result in future work that explores different kinds of cognitive biases and interactions and moves towards more efficient designs.
>
> [1] Kudithipudi, et al. "Biological underpinnings for lifelong learning machines." Nature Machine Intelligence (2022)\
> [2] Hassabis, et al. "Neuroscience-inspired artificial intelligence." Neuron (2017).
>
> > 3 networks are used increasing the computational footprint of the proposed solution. Computational aspect is not discussed. A fair comparison with other methods would be by increasing the buffer size to be equal to the amount of storage due to the extra networks.
>
> The training framework does consist of 3 networks, but however, for inference, still a single network is used. Hence the inference computations remain the same as other methods. However, as mentioned, the computational footprint is higher during training. We would like to put across our thought process regarding this question. Our goal, similar to other method, was to reduce forgetting in networks and improve generalization. Inspired by cognitive theories, we explored the incorporation of multiple elements of cognitive architectures (multi-module approach, cognitive bias and dual memory system) into neural networks to assess the overall gain. All the existing methods have their own specific formulation and tips and tricks to improve continual learning performance. For instance, DER++ stores logits in the buffer, whereas ER does not, and CLS-ER utilizes two extra semantic memories during training, which other methods do not. But, as the goal of all these techniques is to evaluate the CL performance for a specific memory budget, they are all still compared against each other.
>
> Moreover, it is pertinent for a CL technique to show good performance even with limited data and CCL shows gain in lower buffer size setting, also higher than CLS-ER, which also has 3 forward passes(similar to CCL) during training.
> To ensure transparency, we write about the computational complexity aspect in Section F in Appendix.
>
> > The method requires large number of hyperparameters
> We apologize for the lack of information and confusion about the hyperparameters in the paper. To rectify this, we have made a few changes.
> - We have updated the equations (and algorithm) and variable names to ensure better lucidity.
> - A detailed explanation about the different hyperparameters and the search strategies (for baselines and our method) are provided in Section E in Appendix.
> - The learning rate, number of epochs, batch size are consistent across datasets. The additional stochastic update rate and decay factor are similar to parameters in CLS-ER. The loss balancing weights are stable and same for all datasets across all settings. There is also an evident pattern or trend with the dataset complexity. The stochastic update rate is higher for datasets with lesser classes and drops for complex datasets. Overall, the  hyperparameters are quite stable across datasets and settings and further, they also complement each other such that some parameters can be fixed, and the rest can be fine-tuned without much change in performance, thus resulting in lesser search space. Kindly check the edits in the paper and see if it clarifies your concern

---

> > ### Author Response · Authors · 2022-11-13
> > **Reply to Reviewer BtBz (2/3)**
> >
> > >Not sure why the new domain incremental sequence is introduced. Core50 domain incremental learning sequence could be used.
> >
> > Thank you for the question, we would like to elaborate on our reasoning.
> > Core50 is a dataset that was developed for continuous object recognition and has been predominantly used for Class-IL. But it also has the New instance (NI) category that can be used for the domain-incremental setup. We considered this dataset but found some shortcomings, which made us curate a newer dataset for domain incremental learning setup.
> > - Core50 consists of objects that are all handheld by an operator and the point-of-view is from the same operator. This reflects a very specific and simulated/laboratory-generated use case and can not reflect the general real-world settings. There are 50 objects belonging to 10 domestic objects, namely, plug adapters, mobile phones, scissors, light bulbs, cans, glasses, balls, markers, cups and remote controls. These again only represent one category of household or utility items, and the object sizes are all small and pretty similar (as they need to be held in the hand).
> > - The domains are characterized by different backgrounds (indoor, outdoor, different background and lighting conditions). This does not represent a diverse and challenging domain shift. For instance, corruptions such as brightness, contrast, and fog can already simulate such conditions for evaluation.
> >
> > On the other hand, with Domain2IL, we wanted to target multiple functionalities, not just for this work but to be useful for the community in the long run.
> > - The dataset consists of a wide range of categories and objects present in the world, ranging from fauna, flora, vehicles (on roads and in the sky), nature, man-made objects, objects used inside the house, tools used outside, etc. The object sizes vary, are captured in a real-world setting, and reflect a more realistic object recognition problem.
> > - The domain-shift is the most crucial part of choosing to spend resources on using this dataset available for CL domain-IL setting. The six different domains are quite distinct from each other. The change is not in lighting or background but represents a more core distribution shift. Humans can easily identify a train, even if it is in the form of a painting, a billboard or advertisement (infographic), or even a hand-drawn sketch (Figure S4). However, neural networks struggle to adapt between these domains; for instance, check the dip in infographic settings in Table S6. Hence, our goal was to use cognitive-inspired design to assess how well  it can perform on this challenging shift.
> > - The dataset can also be used to perform out-of-distribution analyses on other datasets. We curated the dataset to have similar classes to some of the standard datasets (such as CIFAR10, CIFAR100, and TinyImageNet). Thus, OOD performance can be evaluated by training on any of these datasets and testing on the rest (similar classes).
> > The intention was to curate another dataset that can be used in the CL setting by the whole community for domain-IL, OOD, and performance evaluation and to test the algorithms in a more realistic and challenging data regime.
> >
> > > Only cifar based sequences are considered, what about ImageNet, TinyImageNet or Core50?
> >
> > Our selection of dataset was to include datasets of varying complexity. CIFAR10 was considered to be the simple one and CIFAR100 was the moderate dataset. For a complex dataset, we focused more on the Domain^2IL dataset. Domain^2IL represents a challenging dataset with six diverse domains for Domain-IL setting. Hence, we did not focus on TinyImageNet or ImageNet, as they were also resource intensive.
> >
> > > Not sure why the authors picked only 5 tasks sequence, usually evaluation is done with different number of classes per task in order to show the robustness of the method.
> > Thanks for the suggestion. We scheduled runs on CIFAR100 for more number of tasks and have reported in in Table S2. Due to time constraint, currently, we have reported results for tasks=10 and we report numbers for ER and DER++ (to start the engagement with the reviewers early). Please note that we could not do hyperparameter tuning for CCL, and hence there might be a better setting with higher results. However, we still see gain against all the methods.
> >
> > | B=500 | Method | Task = 5 | Task = 10 |
> > |-------|--------|----------|-----------|
> > |       | ER     | 28.07 ± 0.31    |   21.49±0.47       |
> > |       | DER++  |     41.40±0.96      |  36.20 ±0.52        |
> > |       | CCL    |  **53.23±1.62**       |     **41.09±0.72**     |

---

> > > ### Author Response · Authors · 2022-11-13
> > > **Reply to Reviewer BtBz (3/3)**
> > >
> > > > Not sure that the method is state of the art, there seem to be new published works that show better performance, e.g., PoDNet[1]
> > >
> > > Thanks for the suggestion. We tried to cover different kinds of methods to compare against. ER and DER, which are standard comparisons, CO2L, which involved contrastive objectives that are familiar in self-supervised learning technique, ER-ACE which utilizes metric learning based objectives and CLS-ER which include more than one network for training. Further, we selected the methods that used the standard CL settings that are common through CL literature and those that have reported on multiple different datasets.
> > >
> > > Quoting from the PoDNet paper “we start by training the models on half the classes (i.e., 50 for CIFAR100 and ImageNet100, and 500 for ImageNet1000). Then the classes are added incrementally in steps. ”
> > > The first task trains on half the dataset, which is not the standard setting used in most of CL literature and hence won’t provide a fair comparison to other techniques. Therefore, we did not choose some of the CL methods. However, if this explanation is not sufficient, we would be willing to consider other suggestions for comparison.
> > >
> > >
> > >
> > > > The use of the shape limits the applicability of the method on recognizing objects with discriminating characteristics other than shape, e.g., a horse and a zerbra. [1] Ashok, Arjun, K. J. Joseph, and Vineeth Balasubramanian. "Class-Incremental Learning with Cross-Space Clustering and Controlled Transfer." ECCV 2022.
> > >
> > > We agree with this reasoning. Usage of shape also has limitations , as mentioned by the example by the reviewer.
> > > Hence, we did not want the network to completely rely on shape either. To reduce texture bias and the tendency of networks to learn local cues, we intended the shape to provide subtle supervision to generate better features. A network completely dependent on texture or shape is indeed not an optimal approach. Thus, we wanted to propose a network that learns from both to produce better feature representations. Shape and texture are already present in the data, however networks tend to rely more on texture and less on the global semantics. CCL learns texture and also emphasized on the implicit shape information to generate high level representations that makes use of both the features
> > > In CCL, the knowledge sharing is mutual between texture and shape. So, while one is lagging, the other can help.
> > >
> > > Regarding the "clarity, Quality, Novelty And Reproducibility", we hope the modified section for hyper-parameters ( Section E) and the new tables offer more coherence. Please let us know if we missed any points or if there are more questions and we would be glad to engage. Thanks.

---

> > > > ### Author Response · Authors · 2022-11-18
> > > > **Reply to Reviewer BtBz (4/3)**
> > > >
> > > > We have further added results for 20 tasks for Seq-CIFAR100 in the paper (Table S2 in appendix) and here also for reference.
> > > > CCL outperforms both ER and DER++ (results taken from the existing paper for efficiency) on all the tasks.
> > > >
> > > > | B=500 | Method | Task = 5 | Task = 10 | Task = 20|
> > > > |-------|--------|----------|-----------|-----------|
> > > > |       | ER     | 28.07 ± 0.31    |   21.49±0.47       | 16.52 ±0.86 |
> > > > |       | DER++  |     41.40±0.96      |  36.20 ±0.52        | 22.25 ±5.87 |
> > > > |       | CCL    |  **53.23±1.62**       |     **41.09±0.72**     | **33.60 ±0.25** |

---

> ### Author Response · Authors · 2022-11-18
> **General Response to Reviewer BtBz**
>
> As the first stage of discussion is nearing the deadline, we would like to take the opportunity to check if we have addressed all the concerns and questions. We would be happy to engage more and provide more details if necessary. Thank you.

---

> > ### Comment · Reviewer_BtBz · 2022-11-21
> > **Comment on authors response**
> >
> > I thank the authors for the detailed responses and the efforts they made. I agree with most of the points the authors have mentioned in their response.
> > Regarding the point I raised on the computational cost, the replay buffer is also usually used only during training time with no overhead at test time. Hence, I still think a fair comparison with other methods should be using the same memory footprint.
> > It is interesting to see the rationale behind the chosen datasets but that doesn't necessary exclude other datasets.
> > Section E in the appendix:  "For the other datasets, we ran a grid search over the hyperparameters reported
> > in the paper for a similar dataset". I don't understand what exactly this sentence means.
> > When tuning the hyper-parameters based on validation set, does that mean a validation set from all tasks with the CL performance being the picking criteria?
> > Section E needs revision in terms of writing.

---

> > > ### Author Response · Authors · 2022-11-22
> > > **Response to Reviewer BtBz**
> > >
> > > We would like to thank the reviewer for their encouraging response and the constructive feedback.
> > >
> > > * Computational Cost - We agree with the reviewer that CCL has higher memory footprint during training compared to few of the other techniques reported in the paper. We attempt to further elucidate the comparisons.
> > >   * According to this criterion of having the same computational cost during training, *CLS-ER* and CCL have the same memory footprint (three networks and three forward passes), and hence can be compared against each other without any additional changes.  In this fair comparison, CCL outperforms CLS-ER in all settings as shown in the paper.
> > >   * For all other techniques (*ER, DER++, CO2L, ER-ACE*), which use single network, we can increase the buffer size and compare the results. In Table1, consider CCL results of 200 buffer size for Seq-CIFAR100 (46.55). This is higher than the accuracies of all these four techniques in buffer size 500 (which uses 2.5 times higher memory). Similar results are seen in DomainIL dataset (Figure2/TableS8). To summarize, the CCL results with lower buffer size (200) is higher than the results of the four single-network techniques with higher buffer size (500), and is seen in all versions of Seq-CIFAR100 dataset (GCIL included) and also for the new Domain Incremental datasets (DN4IL).
> > >
> > >    We hope this addresses your concern. Else, please let us know if we can run and provide more results.
> > >
> > > * We agree with inclusion of more datasets, but TinyImageNet and ImageNet datasets are relatively resource and time consuming to obtain the results.
> > > However, if this is the only concern, we can run experiments on TinyImageNet dataset and update the numbers by the camera ready deadline of the paper.
> > >
> > > * Hyperparameters - Thanks for the feedback. We will refine Section E to be more detailed and clear, by including the following:
> > >
> > >   * For every dataset, we utilize a small validation set from the training set to tune the hyperparameters. This validation set is sampled uniformly from each task and is fixed per dataset so all the hyperparameter experiments would use the same set. We follow the continual learning evaluation and consider the final average accuracy on the validation set as the criterion. Once we find the right hyper parameters (with the highest validation set accuracy), we train on all the training data and report the test set accuracy.
> > >   * Regarding this line, "For the other datasets, we ran a grid search over the hyperparameters reported in the paper for a similar dataset". For Seq-CIFAR100, we considered Seq-CIFAR10 parameters (that are reported in paper,) as a reference to create the ranges for grid-search. Similarly, DN4IL dataset is more complex and hence, we consider the Seq-TinyImagenet hyperparameters in the respective paper as the reference point. (For instance, stochastic update rate (r) is much lower for Tinyimagenet (as reported in CLS-ER), hence when finding parameter for DN4IL dataset, the range of r also includes lower values while performing the search).
> > >
> > > Please let us know if this information clarifies your concerns. We can provide more details/results if necessary.

---

> > > > ### Author Response · Authors · 2022-12-08
> > > > **TinyImageNet results added**
> > > >
> > > > Based on the feedback, we started the training on TinyImageNet dataset and we have reported the results in the general comment above. Please let us know if there are more concerns or questions and we will address them.

---

### Official Review · Reviewer_e4pE · 2022-10-25

**Confidence:** 4
**Correctness:** 2
**Technical Novelty And Significance:** 3
**Empirical Novelty And Significance:** 2
**Recommendation:** 6

**Clarity, Quality, Novelty And Reproducibility:**

I have some concerns for this paper regarding reproducibility, in particular with regards to the hyperparameter selection. It is stated in the paper that hyperparameter selection is performed, but it is not reported what values were explored and how they were explored. Moreover, for the proposed CCL method the selected hyperparameters are reported in the Appendix, but for the methods that are compared against no such hyperparameters are reported. In addition, it is not explained how the authors ensured that the hyperparameter selection procedure for the methods that are compared against was done in a way similar to their proposed method.

Code is also not provided to the reviewers, and a proper check for reproducibility issues is therefore not possible.

The clarity of the paper is generally good. I think the paper also has a clear novel component in the proposed use of shape information in continual learning (although the empirical comparisons are not fairly performed as a result of that, see above) and the proposed architecture.


**Strength And Weaknesses:**

The writing and organization of the paper are generally clear, and I find the proposed multi-module architecture an interesting contribution. The Domain2-IL dataset protocol that is constructed could also be a useful contribution (although I’d recommend the authors to reconsider the name).

The paper however has some important issues that require addressing.

First, the fairness of the empirical comparisons. The main empirical component of the paper is comparisons against methods such as ER, DER++ or CLS-ER. However, an important confound in this comparison is that the proposed CCL method uses shape information while the other methods do not. That is, the proposed CCL method uses additional information (or you could see it as additional pre-processing or augmentations) that is not available to the other methods, making these comparisons unfair.
Rather than including these comparisons and trying to make a claim of “state-of-the-art”, I would recommend the authors to instead focus on carefully and systematically characterizing the performance of their proposed methods and its different components (e.g., along the lines of the ablation experiments in Table 2, but more extensively).

Another important issue is the way this paper describes the “continual learning settings”, and its claim that the proposed CCL shows improvement “across all continual learning settings”. By “all continual learning settings” this paper seems to mean Task-IL, Domain-IL, Class-IL and General Class-IL. The distinction between Task-IL, Domain-IL and Class-IL seems based on this paper (https://arxiv.org/abs/1904.07734, although the paper is not discussed or cited, which it probably should be). With regards to these three scenarios I could agree with referring to them as “all settings” (because these are the three ways in which a given sequence of tasks can be performed), but the addition of General Class-IL complicates things. General Class-IL is not just about the way a sequence of tasks is performed, but additionally has restrictions about how that sequence is constructed (e.g., gradual shifts between tasks). When also considering the way a task sequence is constructed, there can be many more settings than just the ones considered in this paper.

It is of course fine to consider a continual learning set up as described by General Class-IL, but it is important not to contrast it (or put it at the same level of abstraction) with Task-IL, Domain-IL and Class-IL. Rather, General Class-IL is a subset of Class-IL, and I think it is important to be clear about that.

As a final note, it might be good to reconsider the name of the proposed Domain2-IL protocol. Currently this name has the risk of suggesting that it is a modified version of the Domain-IL scenario, while it is an instantiation of the Domain-IL scenario (i.e., similar to for example permuted MNIST, but – as the authors rightfully point out – a more realistic one).


**Summary Of The Paper:**

This paper draws inspiration from the cognitive science literature to design a multi-module architecture for continual learning from images. More specifically, the proposed “Cognitive Continual Learning” consists of a working model (which processes images in the typical way), an inductive bias learner (which receives pre-processed shape information) and a semantic memory (which is a delayed version of the working model), whereby the predictions of the different models are encouraged to be similar to each other by various additional loss terms. In extensive empirical comparisons the authors show that the resulting model performs better than several established continual learning methods (e.g., DER++, Co2L and CLS-ER) on variants of task-, domain- and class-incremental learning.


**Summary Of The Review:**

Although I find the proposed multi-module architecture interesting, I believe the provided empirical comparisons are not fair. There is also an important issue regarding the paper’s description of continual learning settings.

---

> ### Author Response · Authors · 2022-11-13
> **Reply to Reviewer e4pE (1/3)**
>
> We would like to thank the reviewer for their feedback and questions. We are glad the reviewer found out multi-module method an interesting contribution. We have addressed the questions below.
>
> > First, the fairness of the empirical comparisons. The main empirical component of the paper is comparisons against methods such as ER, DER++ or CLS-ER. However, an important confound in this comparison is that the proposed CCL method uses shape information while the other methods do not. That is, the proposed CCL method uses additional information (or you could see it as additional pre-processing or augmentations) that is not available to the other methods, making these comparisons unfair. Rather than including these comparisons and trying to make a claim of “state-of-the-art”, I would recommend the authors to instead focus on carefully and systematically characterizing the performance of their proposed methods and its different components (e.g., along the lines of the ablation experiments in Table 2, but more extensively).
>
> We apologize for the lack of clarity on this topic. We would like to clarify that our method does not explicitly use extra/additional information. Texture and shape are both present in the original image, and all networks (even the other methods) have access to all of it. But NNs tend to rely more on texture and ignore semantic information. Hence we utilize the existing shape information to enable the network to also focus on the global shapes while learning. We do not use any generative models or additional data for this task. Also, the way the shape bias is incorporated into the learning is by using a second module (network) that offers supervision throughout the learning process. This is not the same as appending a shape filter to the existing augmentations and feeding the outputs to the same network for learning both RGB and shape images. Such approaches tend to provide sub-optimal representations, as both are of different distributions. Our goal was to allow the network receiving original data to learn on its own modality, while also allowing some supervision from the shape data to help reduce its bias toward texture and local spurious cues and generate better representations.
>
> We want to further clarify the reason for the design and the comparisons. Utilizing the shape information as one of the inductive biases is part of the design of the CCL architecture. We considered the human brain, or cognition, as the most intelligent agent and wanted to incorporate some of the underlying workings into the neural network architecture. Hence, our goal was to devise a framework that is inspired by elements of cognitive architecture and that reduces forgetting while exhibiting better generalization and robustness. Therefore, we wanted to explore how incorporating shape can help in a continual learning setting, compared to other methods. Moreover, each proposed technique has its own specific methodology and tips and tricks to improve continual learning performance. For instance, DER++ stores logits in the buffer, whereas ER does not, and CLS-ER utilizes two extra networks during training, which other methods do not. But, as the goal of all these methods is to improve CL performance, they are all still compared against each other. CCL utilizes cognitive components to gauge how it fares in CL against other methods.
> Finally, through ablation, we wanted to dissect the framework and shed more light on each component, as shape (IB) is not the only component we were inspired by from the cognitive theories. We have tried to cover different ablation cases. We have updated Table 2 to ensure more clarity, and also added more text to the same.
>
> Please let us know if this addresses your concern. We are open to suggestions on providing more ablations and evaluations.

---

> > ### Author Response · Authors · 2022-11-13
> > **Reply to Reviewer e4pE (2/3)**
> >
> > > Another important issue is the way this paper describes the “continual learning settings”, and its claim that the proposed CCL shows improvement “across all continual learning settings”. By “all continual learning settings” this paper seems to mean Task-IL, Domain-IL, Class-IL and General Class-IL. The distinction between Task-IL, Domain-IL and Class-IL seems based on this paper (https://arxiv.org/abs/1904.07734, although the paper is not discussed or cited, which it probably should be). With regards to these three scenarios I could agree with referring to them as “all settings” (because these are the three ways in which a given sequence of tasks can be performed), but the addition of General Class-IL complicates things. General Class-IL is not just about the way a sequence of tasks is performed, but additionally has restrictions about how that sequence is constructed (e.g., gradual shifts between tasks). When also considering the way a task sequence is constructed, there can be many more settings than just the ones considered in this paper.
> >
> > Thank you for pointing this out. We understand the confusion in the way we have represented and phrased about these different settings.
> >
> > Also, thanks for notifying us of the missing citation; the different settings are based on the same paper and we have cited it in the modified version. Based on the paper, we considered the Class-IL, Task-IL, and Domain-IL settings. The paper suggests that Class-IL is the most complex of the three settings. However, another work [1] highlighted some of the limitations of the Class-IL, as it assumes that the number of classes across different tasks is the same, that the classes do not reappear, and that the samples per class are well balanced. Hence they suggested Generalized Class-IL (GCIL) which overcomes these limitations to introduce a more realistic setting: the number of classes, the appearance of these classes, and their sample size are sampled from a distribution.
> >
> > We intended to report Task-IL, Class-IL, and Domain-IL as the three major settings, and additionally, include GCIL as a supplement to Class-IL. We included GCIL in the same table (Table1) to save space, but we have added a vertical line in Table1 to make a distinction of the setting.) We have rephrased the claim in the abstract to “across different settings." Further, we have made this point more clear and provided more explanation in Section 3 (the main paper) and Section D in Appendix. Please let us know if this clarifies this point.
> >
> > [1] Mi, Fei, et al. "Generalized class incremental learning." CVPR workshop 2020.
> >
> > > As a final note, it might be good to reconsider the name of the proposed Domain2-IL protocol. Currently this name has the risk of suggesting that it is a modified version of the Domain-IL scenario, while it is an instantiation of the Domain-IL scenario (i.e., similar to for example permuted MNIST, but – as the authors rightfully point out – a more realistic one).
> >
> > The name originated because of the usage of DomainNet in Domain-IL,  hence the Domain-for-DomainIL, or the “Domain^2IL”.
> > We also wanted to mention that the suggested dataset is not the complete available DomainNet dataset, but a carefully crafted subset (with fewer but relevant classes and samples) which can be used with ease in CL settings for everyone.
> > We changed the name to make it more clear. Is DN4IL better?
> > However, if the reviewer has any suggestions, we would be happy to consider them.

---

> > > ### Author Response · Authors · 2022-11-13
> > > **Reply to Reviewer e4pE (3/3)**
> > >
> > > > I have some concerns for this paper regarding reproducibility, in particular with regards to the hyperparameter selection. It is stated in the paper that hyperparameter selection is performed, but it is not reported what values were explored and how they were explored. Moreover, for the proposed CCL method the selected hyperparameters are reported in the Appendix, but for the methods that are compared against no such hyperparameters are reported. In addition, it is not explained how the authors ensured that the hyperparameter selection procedure for the methods that are compared against was done in a way similar to their proposed method.
> > >
> > > We apologize for the lack of clarity on this topic. To ensure clarity, we made few changes in the names of the loss balancing hyperparameters and also updated the equations and algorithm. We have added a detailed section to address all these concerns (Section E). Results on Seq-CIFAR10 for all baselines were reported from the paper[2][3][4] while for CIFAR-100, GCIL-CIFAR100 and Domain2IL, we ran all the methods. The procedure of how we selected the parameter grid for fine-tuning baseline and also our method is provided with tables and explanation in Section E. Please let us know if it is more clear now.
> > >
> > > > Code is also not provided to the reviewers, and a proper check for reproducibility issues is therefore not possible.
> > >
> > >  Thanks for this feedback. We were planning to publish the code online after acceptance. Hope the new details helps provide more clarity regarding the reproducibility.
> > >
> > > Kindly let us know if we failed to address any point.
> > >
> > > [2] Pietro Buzzega, Matteo Boschini, Angelo Porrello, Davide Abati, and Simone Calderara. Dark experience for general continual learning: a strong, simple baseline. Advances in neural information processing systems, 33:15920–15930, 2020.
> > > [3] Lucas Caccia, Rahaf Aljundi, Nader Asadi, Tinne Tuytelaars, Joelle Pineau, and Eugene Belilovsky. New insights on reducing abrupt representation change in online continual learning. In International Conference on Learning Representations, 2021.
> > > [4] Hyuntak Cha, Jaeho Lee, and Jinwoo Shin. Co2l: Contrastive continual learning. In Proceedings of the IEEE/CVF International Conference on Computer Vision, pp. 9516–9525, 2021.

---

> > ### Comment · Reviewer_e4pE · 2022-11-15
> > **Response to author rebuttal**
> >
> > Thanks to the authors for their rebuttal, and for the changes to the paper. I think these changes have improved the paper. I have updated my score. I now support acceptance of this paper.
> >
> >
> > Regarding the issue of "additionally using shape information", I accept the authors point that technically this is not truly additional information as it is available in the pixel data that is fed to each of the models.
> >
> > Although at the same time, the shape filter is, I presume, trained on data not seen by the other models. Could this not be seen as additional information feeding into the model? I think it would be good to discuss this point in the paper.
> >
> > Also, given previous work demonstrating the bias of DNNs to texture, it seems not very surprising that using the output of a shape filter as an additional input can provide performance benefits. I think the paper could be strengthened further by demonstrating that the particular way in which the shape information is used by the proposed architecture is more beneficial than other (naive) ways of using this information.

---

> > > ### Author Response · Authors · 2022-11-18
> > > **Reply To reviewer e4pE**
> > >
> > > We would like to thank the reviewer for their quick response and their support for our work. We agree with the suggestion of highlighting the shape extraction technique and also demonstrating our way of including shape into learning mechanisms compared to others.
> > >
> > > To this end, we make the following changes:
> > > - The extraction of shape information is pertinent to our work, and hence we included the details in Section B (in Appendix). The shape filter is a Sobel filter, which is a traditional computer vision operation and does not require any training. It is a fixed 3x3 kernel and
> > >  a convolution operation is performed on the input using this kernel. The sobel kernels/filters for extracting edges in the x and y direction and some visual examples are shown in Section B.
> > > This operation is a quick on-the-fly real-time operation. Hence, there is no saving of images in memory. Algorithm1 has a detailed step by step process of this process.
> > >
> > > - The comparison with other techniques is now added in the extended Related works in Section H.
> > > NNs have a tendency for texture-bias and has been demonstrated in multiple works [1][2]. Several techniques have been introduced to reduce texture bias and improve representations. [1] increases shape bias by adding multiple stylized images along with the original images used for training. Styles of artistic paintings are transferred to ImageNet dataset to create stylized-ImageNet. Style-transfer is performed using adaptive instance stylization (ADaIN) and is an additional offline process. [3] uses generative techniques (InfoGan) to synthesize images that are less biased to texture. [4] also creates an augmented dataset by blending the texture of one image and shape of another image in the training set to create a new image. All of these techniques, then merge both the original and synthesized data to train the network on a bigger dataset.
> > >
> > >    However, training a single network with these different distributions leads to learning sub-optimal representations. This is also shown in the results in [1], where the results dropped and they had to do an additional fine-tuning on the original dataset (without style data). Also, synthesizing and generating new data is expensive and might come with an unaccounted bias.
> > > \
> > >     CCL on the other hand, tries to leverage on the under-utilized implicit shape information with minimal overhead. There is no requirement of additional data, generative networks and the rgb and shape data are not combined together to make one big training dataset. The RGB image is fed to one network and the shape information is learnt by another network (IBL) and the supervision is provided via a knowledge transfer between these two networks and is mutual. Each network has enough flexibility to learn on its own feature while also getting guidance from the other.
> > >
> > > We once again thank the reviewer for their detailed feedback.
> > >
> > > \
> > > [1]Geirhos, Robert, et al. "ImageNet-trained CNNs are biased towards texture; increasing shape bias improves accuracy and robustness." ICLR 2019\
> > > [2]Jo, Jason, and Yoshua Bengio. "Measuring the tendency of cnns to learn surface statistical regularities." (2017).\
> > > [3]Chen, Xi, et al. "Infogan: Interpretable representation learning by information maximizing generative adversarial nets." Advances in neural information processing systems 29 (2016).\
> > > [4]Li, Yingwei, et al. "Shape-Texture Debiased Neural Network Training." International Conference on Learning Representations. 2020.

---

### Official Review · Reviewer_J9Rf · 2022-10-31

**Confidence:** 4
**Correctness:** 3
**Technical Novelty And Significance:** 3
**Empirical Novelty And Significance:** 3
**Recommendation:** 5

**Clarity, Quality, Novelty And Reproducibility:**

Clarity and Quality. The paper is well-written, clear, and easy to follow.
Novelty. While all constituent components are not novel, the proposed system in totality is novel.
Reproducibility. Most details are provided to reproduce the results.

**Strength And Weaknesses:**

Strengths:
+ The proposed system is simple combing several elements from recent work but outperforms all baselines.
+ Extensive evaluations are conducted under numerous continual learning settings.

Weakness:
- CCL was compared with a number of recent baselines but the comparison only focused on classification performance disregarding complexity. How does CCL compare with these methods in terms of memory, complexity, etc?
- Common continual learning metrics such as forward and backward knowledge transfer are not reported.
- No comparison to non experience replay methods.

**Summary Of The Paper:**

This paper proposes Cognitive Continual Learner (CCL) which is composed of three modules. An explicit module that learns from the input and two implicit modules (inductive biases and semantic memories) that share indirect contextual knowledge. CCL is evaluated under a number of continual learning settings, including on a novel benchmark, and compared with recent experience replay continual learning methods, demonstrating favourable performance.

**Summary Of The Review:**

The proposed system is interesting combining a few successful elements from recent literature but empirical evaluations (metrics, baselines) require improvement.

---

> ### Author Response · Authors · 2022-11-13
> **Reply to Reviewer J9Rf**
>
> Thank you for the feedback, we address each question below.
>
> > CCL was compared with a number of recent baselines but the comparison only focused on classification performance disregarding complexity. How does CCL compare with these methods in terms of memory, complexity, etc?
>
> Thank you for pointing this out. Computational complexity is pertinent, and hence we added a Section F in Appendix to write about this aspect. To summarize, while the training involves three networks, the inference, like all the other methods, still utilizes a single network. In our approach, only the Semantic memory (SM) is used for all inference and evaluation purposes. However, for training, CCL is composed of 3 networks similar to CLS-ER (hence has 3 forward passes) and they are computationally similar. Note that CCL shows better performance compared to CLS-ER.
>
> The aim of CCL is to utilize cognitive inspired elements to help enhance lifelong learning. The multi-module workings in the brain is the motivation for the three different modules in our architecture. We hope our work acts as a first step towards cognitive CL architectures and the future works can involve improvements on the training efficiency and computational cost.
>
> > Common continual learning metrics such as forward and backward knowledge transfer are not reported.
>
> Thanks for the suggestion. Forward and backward knowledge transfer are metrics shown in literature and we had intended to include them, but we ran into a quandary. These metrics are estimated from the model checkpoint, after a task is completed, as this checkpoint has the highest accuracy for that particular task. However, this does not hold true to our method and CLS-ER, which utilize the stochastically updated model. In our method, SM module assimilates knowledge from the working model and is updated at different frequencies by the exponential moving average. It achieves highest accuracy on previous tasks while also learning the new tasks. Owing to these reasons, the evaluation of these metrics for CCL and CLS-ER might be misleading while compared with other methods. For example, considering Figure S2, if we use the backward metric formula directly, we get a positive backward transfer for CCL in tasks 3 and 4 (due to SM model achieving high accuracy on previous tasks while also learning new tasks at that stochastic update frequency).
>
> However, one way could be to find the best accuracy of the SM module on a particular task and then finding the difference between this and the final accuracy to find the backward transfer. We have explained this and reported preliminary results on one dataset (Seq-CIFAR10) in Section G in appendix. CCL evidently has better backwards transfer.  We can report on other datasets as well and complete the table.
>
> | Method | Backward  Transfer |
> |:------:|:------------------:|
> |   ER   |       -61.75       |
> |  DER++ |       -33.45       |
> | CLS-ER |       -23.47       |
> |   CCL  |        -6.07       |
>
> > No comparison to non experience replay methods.
>
> ER-based approaches have shown state-of-the-art performance in reducing catastrophic forgetting compared to other approaches. The non-ER based methods have the advantage of not needing additional memory, but they still lag behind in performance compared to ER methods. Hence,  we did not compare with the non ER based methods. However, to gauge the overall picture, we added a few standard non ER methods in Table S1 in the Appendix. We report results (from the papers) for Seq-CIFAR10 for four different non-ER based methods.
>
> Please let us know if we addressed all your concerns. We would be happy to discuss more.

---

> > ### Author Response · Authors · 2022-11-18
> > **General Response to Reviewer J9Rf**
> >
> > As the first stage of discussion is nearing the deadline, we would like to take the opportunity to check if we have addressed all the concerns and questions. We would be happy to engage more and provide more details if necessary. Thank you.

---

> ### Author Response · Authors · 2022-11-30
> **Kind reminder**
>
> Dear reviewer, we wanted to check if our response has addressed all your concerns. We would be happy to discuss and answer any remaining queries. Please let us know if there are any additional questions or suggestions that will help increase your support in the acceptance of our paper.\
> Thank you.

---

### Author Response · Authors · 2022-11-13
**General Reply**

We would like to thank all reviewers for their valuable feedback and suggestions. We have addressed the concerns in the individual responses. We have modified the paper and uploaded the revised version. All the changes are in **Blue** color in the revised paper for easy visibility.

**Common Statement**

Our goal was to incorporate some of the key concepts of cognitive workings in the brain into neural networks. CCL provides a framework to integrate multiple modules of the cognitive architectures, different cognitive biases, and dual memory systems. CCL evaluates the benefits of this design in the continual learning setting in mitigating catastrophic forgetting and improving generalization and robustness. Further, we curate a dataset (from DomainNet)  that is well-crafted for the CL domain incremental setting. The goal was to contribute a dataset that is more realistic and has a challenging distribution shift compared to the existing datasets, and hence help the community in benchmarking different CL techniques. Further, while choosing the classes and the samples, care was taken to include classes similar to other standard datasets, such that this dataset could also be used for OOD evaluations. Regarding the complexity, the training requires three networks; however, for inference, a single network is used, similar to other approaches. The results show a promising step towards cognitive-inspired CL architectures and we hope the future work can move towards increasing efficiency in terms of computational cost and also explore utilizing different kinds of cognitive biases, modules and interactions.

**Changes**
- Changed the name of created dataset from Domain$^2$IL to *DN4IL* (DomainNet for Incremental Learning) to avoid any confusion. But, we are open to suggestions and feedback on the same.
- Updated equations and loss balancing parameter names to be more clear and coherent
- Fixed Figure 2 to use averaged values of plasticity and stability
- Simplified the ablation Table 2
- Added comparisons of non ER methods (Table S1)
- Results of Seq-CIFAR100 on a longer task sequence (Table S2)
- Section B and C: Added algorithm for Shape and modified CCL Algorithm
- Section 3 and Section D: More lucidity on the different CL settings
- Section E: Added more details on hyperparameters for baselines and CCL. Tables S3, S4, S5
- Section F: Added section for computational complexity
- Section G: Added section for other metric (backward transfer or forgetting) - Table S7

---

### Author Response · Authors · 2022-12-08
**Additional TinyImageNet results (updated)**

Dear Reviewers,

Based on the feedback, we started the training on TinyImageNet dataset (as an addition to existing CIFAR10, CIFAR100 and the DomainNet datasets) and we report the results below.

The results are promising and as we wanted to engage in discussion instead of waiting for the results, we report the preliminary results and we will update the results for all seeds (**Edit: updated results for all seeds**). Owing to time constraint, hyperparameter tuning was not performed and it is possible to find better parameters.


| Method     |   Seq-TinyImageNet  |
|------------|:-------------------:|
|            |       Class-IL      |
| JOINT      |      59.99±0.19     |
| SGD        |      7.92±0.26      |
|            |                     |
|            | **Buffer size 200** |
| ER         |      8.49±0.16      |
| A-GEM      |      8.07±.0.08     |
| DER++      |      10.96±1.17     |
| CO2L       |      13.88±0.40     |
| CLS-ER     |      23.47±0.80     |
| CCL (Ours) |      **23.58±0.27**      |
|            |                     |
|            | **Buffer size 500** |
| ER         |      9.99±0.29      |
| A-GEM      |      8.06±0.04      |
| DER++      |      19.38±1.41     |
| CO2L       |      20.12±0.42     |
| CLS-ER     |      31.03±0.56     |
| CCL (Ours) |      **31.67±0.47**      |
|            |                     |

---

### Author Response · Authors · 2022-12-08
**Kind Reminder and Thank you.**

We would like to thank all the reviewers for their detailed feedback and review. We have done our best to address all the concerns. Please let us know if there are any additional questions or suggestions that will help increase your support in the acceptance of our work.
Thank you.

---

### Decision · Program_Chairs · 2023-01-20

**Decision:**

Reject

**Justification For Why Not Higher Score:**

1) Limitation of the chosen approach reduces generality; limitation not stated upfront.

2) Lack of novelty

**Justification For Why Not Lower Score:**

n/a

**Metareview: Summary, Strengths And Weaknesses:**

This paper proposes a cognitive architecture-like model for continual learning (CL) from images. Specifically, the proposed Cognitive Continual Learner (CCL) consists of 3 modules: An explicit module ("working model") that learn to classify the inputs, and two implicit modules (for inductive biases and semantic memories). In addition to standard classification losses, there are also a few additional losses to encourage consistency and discourage deviation from historical predictions. A new custom dataset called Domain2IL is also introduced, which is supposed to be a more balanced subset of the existing DomainNet dataset. In extensive empirical comparisons under a number of different common CL settings, the model performs favorably compared to several existing CL methods.

-- STRENGTHS --

1) Evaluations are robust and extensive, covering numerous CL settings.

2) Empirical results are good, outperforming the baselines.


-- WEAKNESSES --

1) Even though reviewers have been convinced by the authors that the use of shape information is legitimate, in that it's not "unfair extra information", it still stands that the usage of such information limits the proposed model to only work properly on visual data, and hence is not a general CL method. While this is not a fatal flaw, this is a clear limitation that must be made explicit early on. Furthermore, this caveat must be appropriately taken into account if true like-for-like comparisons are to be done.

2) There seems to be a lack of any really novel ideas. This was brought up by multiple reviewers, and not really addressed satisfactorily by the authors. It may well be that the approach was not presented in a compelling and interesting way, hence a related criticism that the model *design* was not well-motivated (the authors only addressed broadly the cognitive motivation, not the specific design). I am personally quite familiar with the cognitive science and cognitive architecture literatures -- I would even say I'm quite sympathetic to such approaches -- and yet I found the presentation of the key ideas to be bland and lacking any seeming novelty.

**Summary Of Ac-Reviewer Meeting:**

While most (3/4) of the reviewer scores were leaning positive (6), they clarified they were still ultimately lukewarm about the paper, and none were real supporters.

The authors had responded comprehensively and convincingly on certain initial strong doubts (e.g. usage of shape information, details of the method), and the reviewers readily admitted their minds were changed positively on these points. Nonetheless, there were remaining doubts, as summarized above. While these were not fatal flaws, the key strengths of robust evaluation and good empirical results did not seem to override the perceived weaknesses of novelty and design, which are really the starting points of any strong paper.